# DiffCPS: Diffusion-based Constrained Policy Search for Offline Reinforcement Learning

## Abstract

Constrained policy search (CPS) is a fundamental problem in offline reinforcement learning, which is generally solved by advantage weighted regression (AWR). However, previous methods may still encounter out-of-distribution actions due to the limited expressivity of Gaussian-based policies. On the other hand, directly applying the state-of-the-art models with distribution expression capabilities (i.e., diffusion models) in the AWR framework is insufficient since AWR requires exact policy probability densities, which is intractable in diffusion models. In this paper, we propose a novel approach called **Diffusion-based Constrained Policy Search (DiffCPS)**, which tackles the diffusion-based constrained policy search without resorting to AWR. The theoretical analysis reveals our key insights by leveraging the action distribution of the diffusion model to eliminate the policy distribution constraint in the CPS and then utilizing the Evidence Lower Bound (ELBO) of diffusion-based policy to approximate the KL constraint. Consequently, DiffCPS admits the high expressivity of diffusion models while circumventing the cumbersome density calculation brought by AWR. Extensive experimental results based on the D4RL benchmark demonstrate the efficacy of our approach. We empirically show that DiffCPS achieves better or at least competitive performance compared to traditional AWR-based baselines as well as recent diffusion-based offline RL methods.

## 1 Introduction

Offline Reinforcement Learning (offline RL) aims to seek an optimal policy without environmental interactions (Fujimoto et al., 2019; Levine et al., 2020). This is compelling for having the potential to transform large-scale datasets into powerful decision-making tools and avoids costly and risky online data collection, which offers significant application prospects in fields such as healthcare (Nie et al., 2021; Tseng et al., 2017) and autopilot (Yurtsever et al., 2020; Rhinehart et al., 2018).

Notwithstanding its promise, applying contemporary off-policy RL algorithms (Lillicrap et al., 2015; Fujimoto et al., 2018; Haarnoja et al., 2018a;b) directly into the offline context presents challenges due to distribution shift (Fujimoto et al., 2019; Levine et al., 2020). Previous methods to mitigate this issue under the model-free offline RL setting generally fall into three categories: 1) value function-based approaches, which implement pessimistic value estimation by assigning low values to out-of-distribution actions (Kumar et al., 2020; Fujimoto et al., 2019), 2) sequential modeling approaches, which casts offline RL as a sequence generation task with return guidance (Chen et al., 2021; Janner et al., 2022; Liang et al., 2023; Ajay et al., 2022), and 3) constrained policy search (CPS) approaches, which regularizes the discrepancy between the learned policy and behavior policy (Peters et al., 2010; Peng et al., 2019; Nair et al., 2020). We focus on the CPS-based offline RL due to its convergence guarantee and outstanding performance in a wide range of tasks.

Prior solutions for CPS (Peters et al., 2010; Peng et al., 2019; Nair et al., 2020) primarily train a parameterized unimodal Gaussian policy through weighted regression. However, recent works (Chen et al., 2022; Hansen-Estruch et al., 2023; Shafiullah et al., 2022) show such unimodal Gaussian models in weighted regression will impair the policy performance due to limited distributional expressivity. For example, if we fit a multi-modal distribution with an unimodal Gaussian distribution,

it will unavoidably result in covering the low-density area between peaks. Intuitively, we can choose a more expressive model to eliminate this issue. Nevertheless, Ghasemipour et al. (2020) shows that VAEs (Kingma & Welling, 2013) in BCQ (Fujimoto et al., 2019) do not align well with the behavior dataset, which will introduce the biases in generated actions. Chen et al. (2022) utilizes the diffusion probabilistic model (Sohl-Dickstein et al., 2015; Ho et al., 2020; Song & Ermon, 2019) to generate actions and select the action through the action evaluation model under the well-studied AWR framework. However, AWR requires an exact probability density of behavior policy, which is still intractable for the diffusion-based one. Alternatively, they leverage Monte Carlo sampling to approximate the probability density of behavior policy, which inevitably introduces estimation biases and increases the cost of inference. According to motivating examples in Section 3.1, we visually demonstrate how these issues are pronounced even on a simple bandit task.

To solve the above issues, we present **Diffusion-based Constrained Policy Search (DiffCPS)** which directly solves the constrained policy search problem with a diffusion-based policy. Thereby, we can solve the limited policy expressivity problem and avoid the sampling errors caused by directly using diffusion in AWR. Our proposed method consists of three key parts: 1) We demonstrate that if we use the diffusion model as our policy, we can eliminate the policy distribution constraint in the CPS through the action distribution of the diffusion model; 2) We transform the CPS problem to a convex optimization problem and solve it via Lagrange dual method. We also prove the equivalence between the Lagrange dual problem and the primal convex optimization problem; 3) We approximate the entropy calculation in the Lagrange dual problem with the ELBO of the diffusion-based policy to circumvent the intractable density calculation. Finally, we solve the Lagrange dual problem with gradient descent.

The main contributions of this paper are as follows: 1) We present DiffCPS, which tackles the diffusion-based constrained policy search without resorting to AWR. Thereby, DiffCPS solves the limited policy expressivity problem while avoiding the cumbersome density calculation brought by AWR. 2) We prove that the policy distribution constraint always holds for diffusion-based policy. 3) We prove that the policy constraint can be approximated through the ELBO of diffusion-based policy. 4) Our experimental results illustrate superior or competitive performance compared to existing offline RL methods in D4RL tasks. Even when compared to other diffusion-based methods, DiffCPS also achieves state-of-the-art performance in D4RL MuJoCo locomotion and AntMaze average score. These outcomes substantiate the effectiveness of our method.

## 2 PRELIMINARIES

**Notation:** In this paper, we use $\mu_\theta(\boldsymbol{a}|\boldsymbol{s})$ or $\mu$ to denote the learned policy with parameter $\theta$ and $\pi_b$ to denote behavior policy that generated the offline data. We use superscripts $i$ to denote diffusion timestep and subscripts $t$ to denote RL trajectory timestep. For instance, $a_t^i$ denotes the $t$-th action in $i$-th diffusion timestep.

### 2.1 CONSTRAINED POLICY SEARCH IN OFFLINE RL

Consider a Markov decision process (MDP): $M = \{S, \mathcal{A}, P, R, \gamma, d_0\}$, with state space $S$, action space $\mathcal{A}$, environment dynamics $\mathcal{P}(\boldsymbol{s}'|\boldsymbol{s}, \boldsymbol{a}) : S \times S \times \mathcal{A} \rightarrow [0, 1]$, reward function $R : S \times \mathcal{A} \rightarrow \mathbb{R}$, discount factor $\gamma \in [0, 1)$, and initial state distribution $d_0$. The action-value or Q-value of policy $\mu$ is defined as $Q^\mu(\boldsymbol{s}_t, \boldsymbol{a}_t) = \mathbb{E}_{\boldsymbol{a}_{t+1}, \boldsymbol{a}_{t+2}, \ldots \sim \mu} \left[ \sum_{j=0}^{\infty} \gamma^j r(\boldsymbol{s}_{t+j}, \boldsymbol{a}_{t+j}) \right]$. Our goal is to get a policy to maximize the cumulative discounted reward $J(\theta) = \int_S d_0(\boldsymbol{s}) Q^\mu(\boldsymbol{s}, \boldsymbol{a}) d\boldsymbol{s}$, where $\rho^\mu(\boldsymbol{s}) = \sum_{t=0}^{\infty} \gamma^t p_\mu(\boldsymbol{s}_t = \boldsymbol{s})$ is the unnormalized discounted state visitation frequencies induced by the policy $\mu$ and $p_\mu(\boldsymbol{s}_t = \boldsymbol{s})$ is the likelihood of the policy being in state $\boldsymbol{s}$ after following $\mu$ for $t$ timesteps (Sutton & Barto, 2018; Peng et al., 2019).

In offline setting (Fujimoto et al., 2019), environmental interaction is not allowed, and a static dataset $\mathcal{D} \triangleq \{(\boldsymbol{s}, \boldsymbol{a}, r, \boldsymbol{s}', \text{done})\}$ is used to learn a policy. To avoid out-of-distribution actions, we need to restrict the learned policy $\mu$ to be not far away from the behavior policy $\pi_b$ by the KL divergence constraint. Prior works (Peters et al., 2010; Peng et al., 2019) formulate the above offline RL opti-

mization problem as a constrained policy search and its standard form is as follows

$$\mu^* = \arg\max_{\mu} J(\mu) = \arg\max_{\mu} \int_{\mathcal{S}} d_0(\boldsymbol{s}) \int_{\mathcal{A}} Q^{\mu}(\boldsymbol{s}, \boldsymbol{a}) d\boldsymbol{a} d\boldsymbol{s}$$

$$s.t. \quad D_{\text{KL}}(\pi_b(\cdot|\boldsymbol{s})\|\mu(\cdot|\boldsymbol{s})) \leq \epsilon, \quad \forall \boldsymbol{s} \tag{1}$$

$$\int_{\boldsymbol{a}} \mu(\boldsymbol{a}|\boldsymbol{s}) d\boldsymbol{a} = 1, \quad \forall \boldsymbol{s},$$

Previous works (Peters et al., 2010; Peng et al., 2019; Nair et al., 2020) solved Equation 1 through KKT conditions and get the optimal policy $\pi^*$:

$$\pi^*(\boldsymbol{a}|\boldsymbol{s}) = \frac{1}{Z(\boldsymbol{s})} \pi_b(\boldsymbol{a}|\boldsymbol{s}) \exp\left(\alpha Q_{\phi}(\boldsymbol{s}, \boldsymbol{a})\right), \tag{2}$$

where $Z(\boldsymbol{s})$ is the partition function. Intuitively we can use Equation 2 to optimize policy $\pi$. However, the behavior policy may be very diverse and hard to model. To avoid modeling the behavior policy, prior works (Peng et al., 2019; Wang et al., 2020; Chen et al., 2020) optimize $\pi^*$ through a parameterized policy $\pi_{\theta}$:

$$\arg\min_{\theta} \quad \mathbb{E}_{\boldsymbol{s} \sim \mathcal{D}^{\mu}} \left[ D_{\text{KL}} \left( \pi^*(\cdot|\boldsymbol{s}) \| \pi_{\theta}(\cdot|\boldsymbol{s}) \right) \right]$$

$$= \arg\max_{\theta} \quad \mathbb{E}_{(\boldsymbol{s}, \boldsymbol{a}) \sim \mathcal{D}^{\mu}} \left[ \frac{1}{Z(\boldsymbol{s})} \log \pi_{\theta}(\boldsymbol{a}|\boldsymbol{s}) \exp\left(\alpha Q_{\phi}(\boldsymbol{s}, \boldsymbol{a})\right) \right]. \tag{3}$$

Equation 3 is known as AWR, with $\exp(\alpha Q_{\phi}(\boldsymbol{s}, \boldsymbol{a}))$ being the regression weights. However, AWR requires the exact probability density of policy, which restricts the use of generative models like diffusion models. In this paper, we directly utilize the diffusion-based policy to address Equation 1. Therefore, our method not only avoids the need for explicit probability densities but also solves the limited policy expressivity problem in AWR.

## 2.2 DIFFUSION PROBABILISTIC MODEL

Diffusion model (Sohl-Dickstein et al., 2015; Ho et al., 2020; Song & Ermon, 2019) is a new type of generative model, which has achieved SOTA results in image generation, outperforming other generative models like GAN (Goodfellow et al., 2020; Dhariwal & Nichol, 2021), VAE(Kingma & Welling, 2013) and Flow-based models (Rezende & Mohamed, 2015). Diffusion models are composed of two processes: the forward diffusion process and the reverse process. In the forward diffusion process, we gradually add Gaussian noise to the data $\boldsymbol{x}_0 \sim q(\boldsymbol{x}_0)$ in $T$ steps. The step sizes are controlled by a variance schedule $\beta_i$:

$$q(\boldsymbol{x}_{1:T} \,|\, \boldsymbol{x}_0) := \prod_{i=1}^{T} q(\boldsymbol{x}_i \,|\, \boldsymbol{x}_{i-1}), \quad q(\boldsymbol{x}_i \,|\, \boldsymbol{x}_{i-1}) := \mathcal{N}(\boldsymbol{x}_i; \sqrt{1-\beta_i}\boldsymbol{x}_{i-1}, \beta_i \boldsymbol{I}). \tag{4}$$

In the reverse process, we can recreate the true sample $\boldsymbol{x}_0$ through $p(\boldsymbol{x}^{i-1}|\boldsymbol{x}^i)$:

$$p(\boldsymbol{x}) = \int p(\boldsymbol{x}^{0:T}) d\boldsymbol{x}^{1:T} = \int \mathcal{N}(\boldsymbol{x}^T; \boldsymbol{0}, \boldsymbol{I}) \prod_{i=1}^{T} p(\boldsymbol{x}^{i-1}|\boldsymbol{x}^i) d\boldsymbol{x}^{1:T}. \tag{5}$$

The training objective is to maximize the ELBO of $\mathbb{E}_{\boldsymbol{q}_{\boldsymbol{x}_0}}[\log p(\boldsymbol{x}_0)]$. Following DDPM (Ho et al., 2020), we use the simplified surrogate loss $\mathcal{L}_d(\theta) = \mathbb{E}_{i \sim [1,T], \epsilon \sim \mathcal{N}(\boldsymbol{0}, \boldsymbol{I}), \boldsymbol{x}_0 \sim q} \left[ \|\epsilon - \epsilon_{\theta}(\boldsymbol{x}_i, i)\|^2 \right]$ to approximate the ELBO. After training, sampling from the diffusion model is equivalent to running the reverse process.

## 2.3 CONDITIONAL DIFFUSION PROBABILISTIC MODEL

There are two kinds of conditioning methods: classifier-guided (Dhariwal & Nichol, 2021) and classifier-free (Ho & Salimans, 2021). The former requires training a classifier on noisy data $\boldsymbol{x}_i$ and using gradients $\nabla_{\boldsymbol{x}} \log f_{\phi}(\boldsymbol{y}|\boldsymbol{x}_i)$ to guide the diffusion sample toward the conditioning information $\boldsymbol{y}$. The latter does not train an independent $f_{\phi}$ but combines a conditional noise model $\epsilon_{\theta}(\boldsymbol{x}_i, i, \boldsymbol{s})$ and an unconditional model $\epsilon_{\theta}(\boldsymbol{x}_i, i)$ for the noise. The perturbed noise $w\epsilon_{\theta}(\boldsymbol{x}_i, i) + (w+1)\epsilon_{\theta}(\boldsymbol{x}_i, i, \boldsymbol{s})$ is used to later generate samples. However Pearce et al. (2023) shows this combination will degrade the policy performance in offline RL. Following Pearce et al. (2023); Wang et al. (2022) we solely employ a conditional noise model $\epsilon_{\theta}(\boldsymbol{x}_i, i, \boldsymbol{s})$ to construct our noise model ($w = 0$).

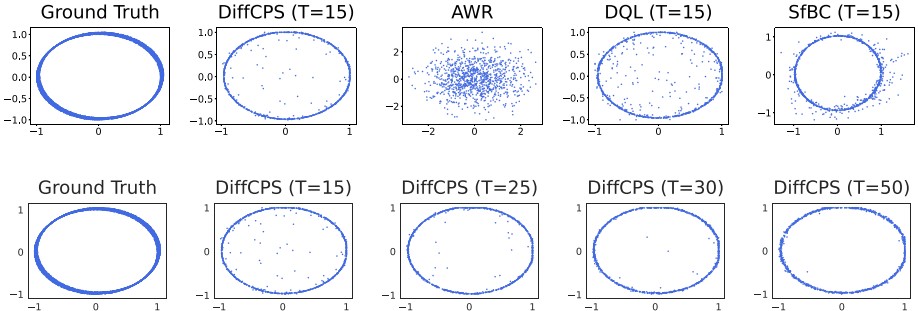

Figure 1: Toy offline experiment on a simple bandit task. We test the performance of AWR and other diffusion-based offline RL algorithms (DQL (Wang et al., 2022) and SfBC (Chen et al., 2022)). The first row displays the actions taken by the trained policy where $T$ denotes diffusion steps. We note that the AWR fails to capture the multi-modal actions in the offline dataset due to the limited policy expressivity of unimodal Gaussian. The second row shows the effect of different diffusion steps $T$.

## 3 METHOD

We propose Diffusion-based Constrained Policy Search (DiffCPS) to address the limited expressivity problem in AWR. Below we first show that the limited policy expressivity in the AWR will degrade the policy performance through a bandit toy experiment. Next, we demonstrate that the sampling errors caused by directly using diffusion within the AWR framework will degrade the performance of the diffusion model. Then we derive our solution to solve this problem from the diffusion model perspective without resorting to AWR. (All proofs are given in Appendix A )

### 3.1 TOY EXPERIMENT

Before showing our method, we first present that the limited policy expressivity in previous Advantage Weighted Regression (AWR) (Peters et al., 2010; Peng et al., 2019; Nair et al., 2020) methods may degrade the performance through a sample 2-D bandit toy experiment with real-valued actions. Our offline dataset is constructed by a unit circle with noise (The first panel of Figure 1). Data on the noisy circle have a positive reward of 1. Note that this offline dataset exhibits strong multi-modality since there are many actions (points on the noisy circle) that can achieve the same reward of 1 if a state is given. However, if we use unimodal Gaussian to represent the policy, although the points on the noisy circle can all receive the same reward of 1, the actions taken by AWR will move closer to the center of the noisy circle (AWR in Figure 1), due to the incorrect unimodal assumption. Experiment results in Figure 1 illustrate that AWR performs poorly in the bandit experiments compared to other diffusion-based methods. We also notice that compared to other diffusion-based methods, actions generated by SfBC include many points that differ significantly from those in the dataset, when $T = 15$. This is due to the sampling error introduced by incorporating diffusion into AWR.

Therefore, we conclude that policy expressivity is important in offline RL since most offline RL datasets are collected from multiple behavior policies or human experts, which exhibit strong multi-modality. To better model behavior policies, we need more expressive generative models to model the policy distribution, rather than using unimodal Gaussians. Furthermore, we also need to avoid the sampling errors caused by using diffusion in AWR, which is the motivation behind our algorithm.

### 3.2 APPLY DIFFUSION BASED POLICY IN CPS

Firstly, we present the form of constrained policy search:

$$\mu^* = \arg\max_{\mu} J(\mu) = \arg\max_{\mu} \int_{\mathcal{S}} d_0(\boldsymbol{s}) \int_{\mathcal{A}} Q^{\mu}(\boldsymbol{s}, \boldsymbol{a}) d\boldsymbol{a} d\boldsymbol{s} \tag{6}$$

$$s.t. \quad E_{\boldsymbol{s} \sim \rho^{\pi_b}(\boldsymbol{s})} \left[ D_{\mathrm{KL}}(\pi_b(\cdot|\boldsymbol{s}) \| \mu(\cdot|\boldsymbol{s})) \right] \leq \epsilon, \tag{7}$$

$$\int_{\boldsymbol{a}} \mu(\boldsymbol{a}|\boldsymbol{s}) d\boldsymbol{a} = 1, \quad \forall \boldsymbol{s}, \tag{8}$$

where $\mu(\boldsymbol{a}|\boldsymbol{s})$ denotes our diffusion-based policy, $\pi_b$ denotes the behavior policy. Here we represent our policy $\mu(\boldsymbol{a}|\boldsymbol{s})$ via the conditional diffusion model:

$$\mu(\boldsymbol{a}|\boldsymbol{s}) = \int \mu(\boldsymbol{a}^{0:T}|\boldsymbol{s})d\boldsymbol{a}^{1:T} = \int \mathcal{N}(\boldsymbol{a}^T; \boldsymbol{0}, \boldsymbol{I}) \prod_{i=1}^{T} \mu(\boldsymbol{a}^{i-1}|\boldsymbol{a}^i, \boldsymbol{s})d\boldsymbol{a}^{1:T}, \tag{9}$$

where the end sample of the reverse chain $\boldsymbol{a}^0$ is the action used for the policy's output and $\mu(\boldsymbol{a}^{0:T})$ is the joint distribution of all noisy samples. According to DDPM, we can approximate the reverse process $\mu(\boldsymbol{a}^{i-1}|\boldsymbol{a}^i, \boldsymbol{s})$ with a Gaussian distribution $\mathcal{N}(\boldsymbol{a}^{i-1}; \boldsymbol{\mu}_\theta(\boldsymbol{a}^i, \boldsymbol{s}, i), \boldsymbol{\Sigma}_\theta(\boldsymbol{a}^i, \boldsymbol{s}, i))$. The training of the diffusion model needs a dataset $\mathcal{D} \sim \mu(\boldsymbol{a}|\boldsymbol{s})$, which is intractable in practice. However, the KL constraint in Equation 7 allows us to train the $\mu(\boldsymbol{a}|\boldsymbol{s})$ with $\mathcal{D} \sim \pi_b(\boldsymbol{a}|\boldsymbol{s})$ because the difference between two policies is small. We also follow the DDPM to fix the covariance matrix and predict the mean with a conditional noise model $\epsilon_\theta(\boldsymbol{a}^i, \boldsymbol{s}, i)$:

$$\boldsymbol{\mu}_\theta(\boldsymbol{a}^i, \boldsymbol{s}, i) = \frac{1}{\sqrt{\alpha_i}}\left(\boldsymbol{a}^i - \frac{\beta_i}{\sqrt{1-\bar{\alpha}_i}}\epsilon_\theta(\boldsymbol{a}^i, \boldsymbol{s}, i)\right). \tag{10}$$

During the reverse process, $\boldsymbol{a}^T \sim \mathcal{N}(\boldsymbol{0}, \boldsymbol{I})$ and then follow the reverse diffusion chain parameterized by $\theta$ as

$$\mathcal{N}\left\{\boldsymbol{a}^{i-1}|\boldsymbol{a}^i\right\} \sim \mathcal{N}\left\{a^{i-1}; \frac{\boldsymbol{a}^i}{\sqrt{\alpha_i}} - \frac{\beta_i}{\sqrt{\alpha_i(1-\bar{\alpha}_i)}}\epsilon_\theta(\boldsymbol{a}^i, \boldsymbol{s}, I), \beta_i\right\} \quad \text{for } i = T, \dots, 1. \tag{11}$$

The reverse sampling in Equation 11 requires iteratively predicting $\epsilon$ $T$ times. When $T$ is large, it will consume much time during the sampling process. To work with small $T$ ($T = 5$ in our experiment), we follow the previous works (Xiao et al., 2021; Wang et al., 2022) to define

$$\beta_i = 1 - \alpha_i = 1 - e^{-\beta_{\min}(\frac{1}{T}) - 0.5(\beta_{\max} - \beta_{\min})\frac{2i-1}{T^2}},$$

which is a noise schedule obtained under the variance preserving SDE of Song et al. (2020).

**Theorem 3.1.** *Let $\mu(\boldsymbol{a}|\boldsymbol{s})$ be a diffusion-based policy and $\pi_b$ be the behavior policy. Then, we have*

*(1) There exists $\kappa_0 \in \mathcal{R}$ such that $\mathbb{E}_{\boldsymbol{s} \sim \rho^{\pi_b}(\boldsymbol{s})}\left[D_{\mathrm{KL}}(\pi_b(\cdot|\boldsymbol{s})\|\mu(\cdot|\boldsymbol{s}))\right] \leq \epsilon$ can be transformed to*

$$H(\pi_b, \mu) = -\mathbb{E}_{\boldsymbol{s} \sim \rho^{\pi_b}(\boldsymbol{s}), \boldsymbol{a} \sim \pi_b(\cdot|\boldsymbol{s})}\left[\log \mu(\cdot|\boldsymbol{s})\right] \leq \kappa_0,$$

*which is a convex function of $\mu$.*

*(2) $\forall s, \int_{\boldsymbol{a}} \mu(\boldsymbol{a}|\boldsymbol{s})d\boldsymbol{a} \equiv 1$.*

**Corollary 3.1.1.** *The primal problem (Equation 6-Equation 8) can be transformed into the following optimization problem:*

$$\mu^* = \arg\max_\mu J(\mu) = \arg\max_\mu \int_{\mathcal{S}} d_0(\boldsymbol{s}) \int_{\mathcal{A}} Q^\mu(\boldsymbol{s}, \boldsymbol{a})d\boldsymbol{a}d\boldsymbol{s} \tag{12}$$

$$s.t. \quad H(\pi_b, \mu) \leq \kappa_0.$$

### 3.3 DIFFUSION-BASED CONSTRAINED POLICY SEARCH

In this section, the duality is used to solve the Equation 12 and derive our method. The Lagrangian of Equation 12 is

$$\mathcal{L}(\mu, \lambda) = J(\mu) + \lambda(\kappa_0 - H(\pi_b, \mu)). \tag{13}$$

The Lagrange dual problem associated with the Equation 12 is

$$\min_{\lambda \geq 0} \max_\mu J(\mu) + \lambda(\kappa_0 - H(\pi_b, \mu)). \tag{14}$$

**Theorem 3.2.** *Suppose that $r$ is bounded and that Slaters' condition holds for our offline RL setting. Then, strong duality holds for Equation 12, which has the same optimal policy $\mu^*$ with Equation 14.*

According to Theorem 3.2, we can get the optimal policy by solving Equation 14, where $\lambda$ is the dual variable. We can solve the optimal dual variable $\lambda$ as

$$\arg\min_{\lambda \geq 0} \lambda(\kappa_0 - H(\pi_b, \mu^*)), \tag{15}$$

where $\mu^*$ denotes

$$\arg\max_\mu \mathbb{E}_{\boldsymbol{s} \sim d_0(\boldsymbol{s}), \boldsymbol{a} \sim \mu(\cdot|\boldsymbol{s})}\left[Q^\mu(\boldsymbol{s}, \boldsymbol{a})\right] - \lambda H(\pi_b, \mu). \tag{16}$$

In Equation 16, we need to calculate the cross entropy $H(\pi_b, \mu)$, which is intractable in practice.

**Proposition 3.1.** *In the diffusion model, we can approximate the entropy with an MSE-like loss* $\mathcal{L}_c(\pi_b, \mu)$ *through ELBO:*

$$H(\pi_b, \mu) \approx c + \mathcal{L}_c(\pi_b, \mu), \tag{17}$$

*where c is a constant.*

Let $\kappa = \kappa_0 - c$, according to Proposition 3.1, the Equation 16 and Equation 15 can be transformed to

$$\arg\max_{\mu} \mathbb{E}_{\boldsymbol{s} \sim d_0(\boldsymbol{s}), \boldsymbol{a} \sim \mu(\cdot|\boldsymbol{s})} \left[ Q^{\mu}(\boldsymbol{s}, \boldsymbol{a}) \right] - \lambda \mathcal{L}_c(\pi_b, \mu), \tag{18}$$

$$\arg\min_{\lambda \geq 0} \mathcal{J}(\lambda) = \lambda(\kappa - \mathcal{L}_c(\pi_b, \mu^*)). \tag{19}$$

In practice, the $Q^{\mu}(\boldsymbol{s}, \boldsymbol{a})$ varies in different environments. To normalize $Q^{\mu}(\boldsymbol{s}, \boldsymbol{a})$, we follow Fujimoto & Gu (2021); Wang et al. (2022) to divide it by the target Q-net's value. We also clip the $\lambda$ to keep the constraint $\lambda \geq 0$ holding by $\lambda_{\text{clip}} = \max(c, \lambda), c \geq 0$. So actually $\mu^*$ is

$$\arg\max_{\mu} \mathcal{J}(\mu) = \mathbb{E}_{\boldsymbol{s} \sim d_0(\boldsymbol{s}), \boldsymbol{a} \sim \mu(\cdot|\boldsymbol{s})} \left[ \frac{Q^{\mu}(\boldsymbol{s}, \boldsymbol{a})}{Q^{\mu}_{\text{target}}(\boldsymbol{s}, \boldsymbol{a})} \right] - \lambda_{\text{clip}} \mathcal{L}_c(\pi_b, \mu). \tag{20}$$

Theoretically, we need to precisely solve the Equation 20 and Equation 19, but in practice, we can resort to stochastic gradient descent to solve the equations. In this way, we can recursively optimize Equation 12 through Equation 20 and Equation 19. The policy improvement described in Equation 20 and the solution of Equation 19 constitute the core of DiffCPS.

We also find that we can improve the performance of the policy by delaying the policy update (Fujimoto et al., 2018). Finally, we summarize our method in Algorithm 1:

---

**Algorithm 1** DiffCPS

---

Initialize policy network $\mu_\theta$, critic networks $Q_{\phi_1}, Q_{\phi_2}$, and target networks $\mu_{\theta'}, Q_{\phi'_1}, Q_{\phi'_2}$, policy evaluation interval $d$ and step size $\eta$.
**for** $t = 1$ to $T$ **do**
    Sample transition mini-batch $\mathcal{B} = \{(\boldsymbol{s}_t, \boldsymbol{a}_t, r_t, \boldsymbol{s}_{t+1})\} \sim \mathcal{D}$.
    *# Critic updating*
    Sample $\boldsymbol{a}_{t+1} \sim \mu_\theta(\cdot|\boldsymbol{s}_{t+1})$ according to Equation 11.
    $y = r_t + \gamma \min_{i=1,2} Q_{\phi'_i} \{\boldsymbol{s}_{t+1}, \boldsymbol{a}_{t+1}\}$.
    Update critic $\phi_i \leftarrow \arg\min_{\phi_i} N^{-1} \sum (y - Q_{\phi_i}(\boldsymbol{s}_t, \boldsymbol{a}_t))^2$.
    **if** $t \bmod d$ **then**
        Sample $\boldsymbol{a} \sim \mu_\theta(\cdot|\boldsymbol{s})$ according to Equation 11.
        *# Policy updating through Equation 20*
        $\mu_i \leftarrow \mu_i + \eta \nabla_\theta \mathcal{J}(\mu)$.
        *# Lagrange multiplier $\lambda$ updating through Equation 19*
        $\lambda_i \leftarrow \lambda_i - \eta \nabla_\lambda \mathcal{J}(\lambda)$.
        $\lambda_i \leftarrow \lambda_{\text{clip}} = \max(c, \lambda_i), c \geq 0$.
    **end if**
    *# Target Networks updating*
    $\theta' = \rho\theta' + (1-\rho)\theta, \phi'_i = \rho\phi'_i + (1-\rho)\phi_i$ for $i = \{1, 2\}$.
**end for**

---

## 4 EXPERIMENTS

We evaluate our DiffCPS on the D4RL (Fu et al., 2020) benchmark in Section 4.1. Further, we conduct an ablation experiment to assess the contribution of different parts in DiffCPS in Section 4.2.

### 4.1 COMPARISON TO OTHER METHODS

In Table 1, we compare the performance of DiffCPS to other offline RL methods in D4RL (Fu et al., 2020) tasks. We only focus on the MuJoCo locomotion and AntMaze tasks due to the page limit.

| Dataset | Environment | CQL | IDQL-A | QGPO | SfBC | DD | Diffuser | Diffuison-QL | IQL | DiffCPS(ours) |
|---|---|---|---|---|---|---|---|---|---|---|
| Medium-Expert | HalfCheetah | 62.4 | **95.9** | 93.5 | 92.6 | 90.6 | 79.8 | **96.8** | 86.7 | $100.3 \pm 4.1$ |
| Medium-Expert | Hopper | 98.7 | 108.6 | 108.0 | 108.6 | **111.8** | 107.2 | **111.1** | 91.5 | $112.1 \pm 0.6$ |
| Medium-Expert | Walker2d | 110.1 | **112.7** | **110.7** | 109.8 | 108.8 | 108.4 | 110.1 | 109.6 | $113.1 \pm 1.8$ |
| Medium | HalfCheetah | 44.4 | 51.0 | **54.1** | 45.9 | 49.1 | 44.2 | **51.1** | 47.4 | $71.0 \pm 0.5$ |
| Medium | Hopper | 58.0 | 65.4 | **98.0** | 57.1 | 79.3 | 58.5 | **90.5** | 66.3 | $100.1 \pm 3.5$ |
| Medium | Walker2d | 79.2 | 82.5 | **86.0** | 77.9 | 82.5 | 79.7 | 87.0 | 78.3 | $90.9 \pm 1.6$ |
| Medium-Replay | HalfCheetah | 46.2 | 45.9 | **47.6** | 37.1 | 39.3 | 42.2 | **47.8** | 44.2 | $50.5 \pm 0.6$ |
| Medium-Replay | Hopper | 48.6 | 92.1 | 96.9 | 86.2 | **100** | 96.8 | 101.3 | 94.7 | $101.1 \pm 0.2$ |
| Medium-Replay | Walker2d | 26.7 | **85.1** | 84.4 | 65.1 | 75.0 | 61.2 | **95.5** | 73.9 | $91.3 \pm 0.7$ |
| **Average (Locomotion)** | | 63.9 | 82.1 | 86.6 | 75.6 | 81.8 | 75.3 | 87.9 | 76.9 | **92.26** |
| Default | AntMaze-umaze | 74.0 | **94.0** | **96.4** | 92.0 | - | - | 93.4 | 87.5 | $97.4 \pm 3.7$ |
| Diverse | AntMaze-umaze | **84.0** | 80.2 | 74.4 | **85.3** | - | - | 66.2 | 62.2 | $87.4 \pm 3.8$ |
| Play | AntMaze-medium | 61.2 | **84.5** | **83.6** | 81.3 | - | - | 76.6 | 71.2 | $88.2 \pm 2.2$ |
| Diverse | AntMaze-medium | 53.7 | **84.8** | **83.8** | 82.0 | - | - | 78.6 | 70.0 | $87.8 \pm 6.5$ |
| Play | AntMaze-large | 15.8 | **63.5** | **66.6** | 59.3 | - | - | 46.4 | 39.6 | $65.6 \pm 3.6$ |
| Diverse | AntMaze-large | 14.9 | **67.9** | **64.8** | 45.5 | - | - | 57.3 | 47.5 | $63.6 \pm 3.9$ |
| **Average (AntMaze)** | | 50.6 | 79.1 | 78.3 | 74.2 | - | - | 69.8 | 63.0 | **81.67** |
| **# Diffusion steps** | | - | 5 | 15 | 15 | 100 | 100 | 5 | - | 5 |

Table 1: The performance of DiffCPS and other SOTA baselines on D4RL tasks. The mean and standard deviation of DiffCPS are obtained by evaluating the trained policy on five different random seeds. We report the performance of baseline methods using the best results reported from their paper. "-A" refers to any number of hyperparameters allowed. Results within 3 percent of the maximum in every D4RL task and the best average result are highlighted in boldface.

| D4RL Tasks | DiffCPS (T=5) | DiffCPS (T=15) | DiffCPS (T=20) | SfBC (T=10) | SfBC (T=15) | SfBC (T=25) |
|---|---|---|---|---|---|---|
| **Locomotion** | **92.0** | 87.5 | 87.6 | 72.9 | **75.6** | 74.4 |
| **AntMaze** | **80.0** | 60.7 | 66.7 | 65.7 | **74.2** | 73.0 |

Table 2: Ablation study of diffusion steps. We conduct an ablation study to investigate the impact of diffusion steps on different algorithms. We only show the average score due to the page limit. The results of DiffCPS are obtained from three random seeds, while the results of SfBC are derived from the original SfBC paper.

In traditional MuJoCo tasks, DiffCPS outperforms other methods as well as recent diffusion-based method (Wang et al., 2022; Lu et al., 2023; Hansen-Estruch et al., 2023) by large margins in most tasks, especially in the HalfCheetah medium. Note the medium datasets are collected by an online SAC (Haarnoja et al., 2018a) agent trained to approximately $1/3$ the performance of the expert. Hence, the medium datasets contain a lot of suboptimal trajectories, which makes the offline RL algorithms hard to learn.

Compared to the locomotion tasks, the AntMaze tasks are more challenging since the datasets consist of sparse rewards and suboptimal trajectories. Even so, DiffCPS also achieves competitive or SOTA results compared with other methods. In relatively simpler tasks like umaze, DiffCPS can achieve a $100 \pm 0\%$ success rate on some seeds, which shows the powerful ability to learn from suboptimal trajectories of our method. In other AntMaze tasks, DiffCPS also shows competitive performance compared to other state-of-the-art diffusion-based approaches. Overall, DiffCPS outperforms the state-of-the-art algorithms, even the recent diffusion-based state-of-the-art algorithms by a very significant margin.

## 4.2 ABLATION STUDY

In this section, we analyze why DiffCPS outperforms the other methods quantitatively on D4RL tasks. We conduct an ablation study to investigate the impact of three parts in DiffCPS, *i.e.* diffusion steps, the minimum value of Lagrange multiplier $\lambda_{\text{clip}}$, and policy evaluation interval.

**Diffusion Steps**. We show the effect of diffusion steps $T$, which is a vital hyperparameter in all diffusion-based methods. In SfBC the best $T$ is 15, while $T = 5$ is best for our DiffCPS. Table 2 shows the average performance of different diffusion steps. Figure 2 shows the training curve of selected D4RL tasks over different diffusion steps $T$.

We also note that large $T$ works better for the bandit experiment. However, for D4RL tasks, a large $T$ will lead to a performance drop. The reason is that compared to bandit tasks, D4RL datasets contain

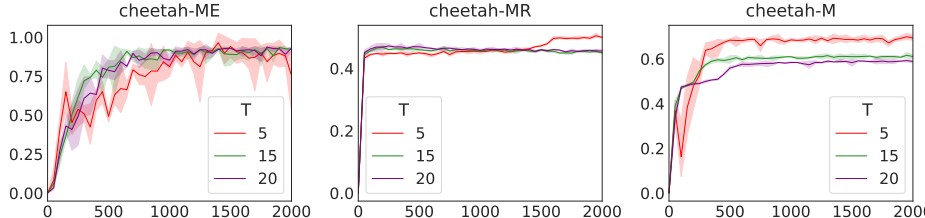

Figure 2: Ablation studies of diffusion steps $T$ on selected Gym tasks (three random seeds). We observe that as $T$ increases, the training stability improves, but the final performance drops.

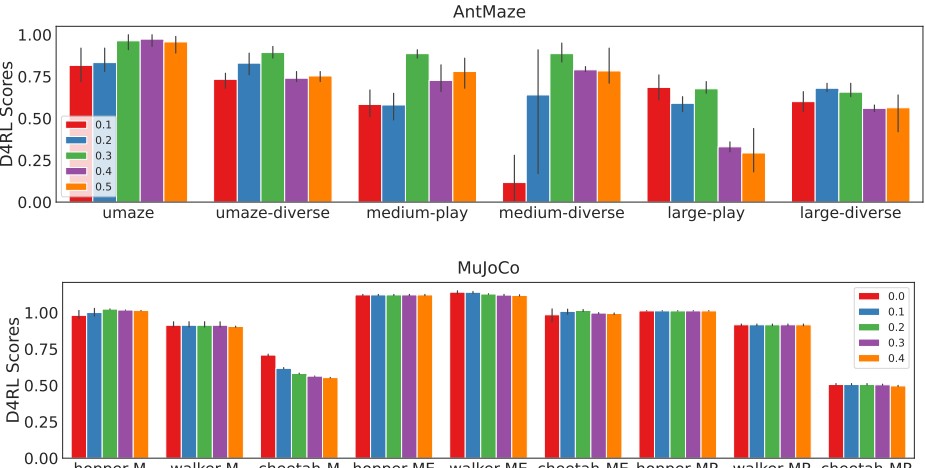

Figure 3: Ablation studies of the $\lambda_{\text{clip}}$ in AntMaze and MuJoCo tasks. We can observe that $\lambda_{\text{clip}}$ has little impact on MuJoCo tasks but significantly influences AntMaze tasks, especially as the AntMaze datasets become larger. The reason is that the sparse rewards and suboptimal trajectories in the AntMaze datasets make the critic network prone to error estimation, leading to learning poor policy. Therefore, there is a need to enhance learning from the original dataset which means we should increase $\lambda$ or enhance the KL constraint. We find that increasing $\lambda_{\text{clip}}$ while maintaining a moderate KL constraint achieves the best results. All the results are obtained by evaluating three random seeds.

a significant amount of suboptimal trajectories. A larger $T$ implies stronger behavior cloning ability, which can indeed lead to policy overfitting to the suboptimal data, especially when combined with actor-critic methods. Poor policies result in error value estimates, and vice versa, creating a vicious cycle that leads to a drop in policy performance with a large $T$.

**The Minimum value of Lagrange multiplier $\lambda_{\text{clip}}$.** In our method, $\lambda$ serves as the coefficient for the policy constraint term, where a larger $\lambda$ implies a stronger policy constraint. Although we need to restrict the $\lambda \geq 0$ according to the definition of Lagrange multiplier, we notice that we could get better results through clip $\lambda \geq c$ in AntMaze tasks, where $c$ is a positive number, see full $\lambda$ ablation results in Figure 3 for details.

**Policy evaluation interval.** We include the ablation of policy evaluation interval in Figure 4. We find that the policy delayed update (Fujimoto et al., 2018) has significant impacts on AntMaze large tasks. However, for other tasks, it does not have much effect and even leads to a slight performance decline. The reason is that infrequent policy updates reduce the variance of value estimates, which is more effective in tasks where sparse rewards and suboptimal trajectories lead to significant errors in value function estimation like AntMaze large tasks.

In a nutshell, the ablation study shows that the combination of these three parts, along with DiffCPS, collectively leads to producing good performance.

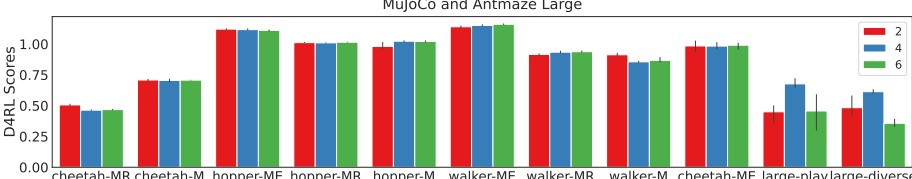

Figure 4: Ablation studies of the policy evaluation interval in AntMaze and MuJoCo tasks. Delayed policy updates have a relatively minor impact on the MuJoCo locomotion tasks. However, for large-scale sparse reward datasets like AntMaze Large, choosing an appropriate update frequency can greatly increase the final optimal results. The MuJoCo task results are obtained with 2 million training steps (three random seeds), while AntMaze results are obtained with 1 million training steps (three random seeds).

## 5 RELATED WORK

**Offline Reinforcement Learning**. Offline RL algorithms need to avoid extrapolation error. Prior works usually solve this problem through policy regularization (Fujimoto et al., 2019; Kumar et al., 2019; Wu et al., 2019; Fujimoto & Gu, 2021), value pessimism about unseen actions (Kumar et al., 2020; Kostrikov et al., 2021a), or implicit TD backups (Kostrikov et al., 2021b; Ma et al., 2021) to avoid the use of out-of-distribution actions. Another line of research solves the offline RL problem through weighted regression (Peters et al., 2010; Peng et al., 2019; Nair et al., 2020) from the perspective of CPS. Our DiffCPS derivation is related but features with a diffusion model form.

**Diffusion models in RL**. Our model introduces the diffusion model to RL. To that end, we review works that use the diffusion model in RL. Diffuser (Janner et al., 2022) uses the diffusion model to directly generate trajectory guided with gradient guidance or reward. DiffusionQL (Wang et al., 2022) uses the diffusion model as an actor and optimizes it through the TD3+BC-style objective with a coefficient $\eta$ to balance the two terms. AdaptDiffuser (Liang et al., 2023) uses a diffusion model to generate extra trajectories and a discriminator to select desired data to add to the training set to enhance the adaptability of the diffusion model. DD (Ajay et al., 2022) uses a conditional diffusion model to generate trajectory and compose skills. Unlike Diffuser, DD diffuses only states and trains inverse dynamics to predict actions. QGPO (Lu et al., 2023) uses the energy function to guide the sampling process and proves that the proposed CEP training method can get an unbiased estimation of the gradient of the energy function under unlimited model capacity and data samples. IDQL (Hansen-Estruch et al., 2023) reinterpret IQL as an Actor-Critic method and extract the policy through sampling from a diffusion-parameterized behavior policy with weights computed from the IQL-style critic. DiffCPS is distinct from these methods because we derive it from CPS.

Closest to our work is the method that combines AWR and diffusion models. SfBC (Chen et al., 2022) uses the diffusion model to generate candidate actions and uses the regression weights to select the best action. Our method differs from it as we directly solve the limited policy expressivity problem through the diffusion-based policy without resorting to AWR. This makes DiffCPS simple to implement and tune hyperparameters. As shown in Table 4 we can only tune one hyperparameter to get SOTA results in most tasks.

## 6 CONCLUSION

In our work, we solve the limited expressivity problem in the weighted regression through the diffusion model. We first simplify the CPS problem with the action distribution of diffusion-based policy. Then we reformulate the CPS problem as a convex optimization problem and solve it by using the Lagrange dual method. Finally, we approximate the entropy with the ELBO of the diffusion model to circumvent the intractable density calculation and approximate solve the Lagrange dual problem by iteratively gradient descent. Experimental results on the D4RL benchmark illustrate the superiority of our method which outperforms previous SOTA algorithms in most tasks, and DiffCPS is easy to tune hyperparameters, which only needs to tune the constraint $\kappa$ in most tasks. We hope that our work can inspire relative researchers to utilize powerful generative models, especially the diffusion model, for offline reinforcement learning and decision-making.

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

# A PROOFS

## A.1 THEOREM 3.1

We provide the proof of Theorem 3.1.

**Proof.** (1): Since the pointwise KL constraint in Equation 1 at all states is intractable, we follow the Peng et al. (2019) to relax the constraint by enforcing it only in expectation

$$E_{s \sim \rho^{\pi_b}(s)} [D_{\mathrm{KL}}(\pi_b(\cdot|s) \| \mu(\cdot|s))] \leq \epsilon.$$

The LHS can be transformed to

$$E_{s \sim \rho^{\pi_b}(s)} [D_{\mathrm{KL}}(\pi_b(\cdot|s) \| \mu(\cdot|s))] = E_{s \sim \rho^{\pi_b}(s), a \sim \pi_b(\cdot|s)} [\log \pi_b(\cdot|s) - \log \mu(\cdot|s)], \quad (21)$$

where

$$E_{s \sim \rho^{\pi_b}(s), a \sim \pi_b(\cdot|s)} [\log \pi_b(\cdot|s)]$$

is the negative entropy of $\pi_b$ is a constant for $\mu$. So the constraint in Equation 7 can be transformed to

$$H(\pi_b, \mu) = -\mathbb{E}_{s \sim \rho^{\pi_b}(s), a \sim \pi_b(\cdot|s)} [\log \mu(\cdot|s)] \leq \kappa_0, \quad (22)$$

where $\kappa_0 = \epsilon - E_{s \sim \rho^{\pi_b}(s), a \sim \pi_b(\cdot|s)} [\log \pi_b(\cdot|s)]$.

(2): According to the definition of the diffusion model in Equation 9:

$$
\begin{aligned}
\int_a \mu(a|s) da &= \int_a \mu(a^0|s) da^0 \\
&= \int \int \mu(a^{0:T}|s) da^{1:T} da^0 \\
&= \int \mu(a^{0:T}) da^{0:T} \\
&= 1.
\end{aligned}
\quad (23)
$$

Equation 23 implies the constraint Equation 8 always holds for the diffusion model, so we can omit this constraint.

According to the above two properties, we can rewrite the diffusion-based CPS problem Equation 6-Equation 8 as

$$
\begin{aligned}
\mu^* = \arg\max_\mu J(\mu) &= \arg\max_\mu \int_S d_0(s) \int_A Q^\mu(s, a) da \, ds \\
s.t. \quad H(\pi_b, \mu) &\leq \kappa_0.
\end{aligned}
\quad (24)
$$

$\square$

## A.2 THEOREM 3.2

Then we prove the Theorem 3.2.

According to Theorem 3.1, the primal problem of DiffCPS is expressed as Equation 24. We first transform the Equation 24 to a more tractable form, and then we prove that the strong duality holds for the simplified Equation 29 via the Fenchel-Moreau perturbation theory. Next, we will transform Equation 24 into a more tractable form. Before we proceed, let's introduce some important concepts.

Defining the occupation measure $\rho^\mu(s, a) = (1 - \gamma) \sum_{t=0}^\infty \gamma^t p_\mu(s_t = s) \mu(a|s)$, it follows that $\rho^\mu(s, a) = (1 - \gamma) \rho^\mu(s) \mu(a|s)$, where $1 - \gamma$ is the normalized factor and $\rho^\mu(s)$ denotes state visitation probabilities with the following properties

$$\rho^\mu(s') = d_0(s') + \gamma \int_{S \times A} \mathcal{T}(s'|s, a) \mu(a|s) \rho^\mu(s) da \, ds \quad (25)$$

Then we can get for all $\mu_1, \mu_2$,

$$
\begin{aligned}
&\mathbb{E}_{s'}\left[|\rho_1(s') - \rho_2(s')|\right] \\
&= \mathbb{E}\left[\gamma\left[\iint_{\mathcal{S}\times\mathcal{A}}\mathcal{T}(s'|s,a)|\mu_1(a|s)\rho_1(s) - \mu_2(a|s)\rho_2(s)|dads\right]\right] \\
&= \frac{\gamma}{1-\gamma}\mathbb{E}\left[\left[\int_{\mathcal{S}\times\mathcal{A}}\mathcal{T}(s'|s,a)|\rho_1(s,a) - \rho_2(s,a)|dads\right]\right] \\
&\leq \frac{\gamma}{1-\gamma}\max_{s,a}\bar{\rho}(s,a),
\end{aligned}
\tag{26}
$$

where $\bar{\rho}(s,a) = |\rho_1(s,a) - \rho_2(s,a)|$.

Because all feasible policies satisfy the KL divergence constraint, when the KL divergence constraint is sufficiently strong, the differences between the occupancy measures for different policies are small. Therefore, based on Equation 26, we can approximately assume that for all feasible policies $\rho^{\pi_b}(s) \approx \rho^{\mu_1}(s) \approx \rho^{\mu_2}(s)$. Thus, Constraint $H(\pi_b, \mu)$ can be simplified to

$$
\begin{aligned}
-H(\pi_b, \mu) &= \mathbb{E}_{s\sim\rho^{\pi_b}(s), a\sim\pi_b(\cdot|s)}\left[\log\mu(\cdot|s)\right] \\
&= \int_{\mathcal{S}\times\mathcal{A}}\rho^{\pi_b}(s)\pi_b(a|s)\log\mu(a|s)dads \\
&= \frac{1}{1-\gamma}\int_{\mathcal{S}\times\mathcal{A}}\rho^{\pi_b}(s,a)\log\frac{\rho^{\mu}(s,a)}{\rho^{\mu}(s)}dads - \underbrace{\frac{1}{1-\gamma}\int_{\mathcal{S}\times\mathcal{A}}\rho^{\pi_b}(s,a)\log(1-\gamma)dads}_{C_0} \\
&\approx \underbrace{\frac{1}{1-\gamma}\int_{\mathcal{S}\times\mathcal{A}}\rho^{\pi_b}(s,a)\log\frac{\rho^{\mu}(s,a)}{\rho^{\pi_b}(s)}dads}_{-\bar{H}(\pi_b,\mu)} - \underbrace{\frac{1}{1-\gamma}\int_{\mathcal{S}\times\mathcal{A}}\rho^{\pi_b}(s,a)\log(1-\gamma)dads}_{C_0},
\end{aligned}
\tag{27}
$$

where $C_0$ is the constant about $\mu$. So we can approximate the constraint in Equation 24 with

$$
-\bar{H}(\pi_b, \mu) \geq -\bar{\kappa}_0,
\tag{28}
$$

where $\bar{\kappa}_0 = \kappa_0 - C_0$. So we can transform the Equation 24 to

$$
\begin{aligned}
\mu^* = \arg\max_\mu J(\mu) &= \arg\max_\mu \int_{\mathcal{S}}d_0(s)\int_{\mathcal{A}}Q^{\mu}(s,a)dads \\
s.t. \quad &-\bar{H}(\pi_b, \mu) \geq -\bar{\kappa}_0.
\end{aligned}
\tag{29}
$$

The perturbation function associated to Equation 29 is defined as

$$
\begin{aligned}
P(\xi) = \max_\mu \mathcal{J}(\mu) &= \max_\mu \int_{\mathcal{S}}d_0(s)\int_{\mathcal{A}}Q^{\mu}(s,a)dads \\
s.t. \quad &-\bar{H}(\pi_b, \mu) \geq -\bar{\kappa}_0 + \xi.
\end{aligned}
\tag{30}
$$

**Lemma A.1.** *If (i) $r$ is bounded; (ii) Slater's condition holds for Equation 29 and (iii) its perturbation function $P(\xi)$ is concave, then strong duality holds for Equation 29.*

**Proof.** See, e.g., Rockafellar (1970)[Cor. 30.2.2] $\qquad\square$

**Proof.** (The proof skeleton is essentially based on Paternain et al. (2019) Theorem 1)

Condition (i) and (ii) are satisfied by the hypotheses of Theorem 3.2. To prove the strong duality of Equation 24, it suffices then to show the perturbation function is concave [(iii)], i.e., that for every $\xi^1, \xi^2 \in \mathbb{R}$, and $t \in (0, 1)$,

$$
P\left[t\xi^1 + (1-t)\xi^2\right] \geq tP\left(\xi^1\right) + (1-t)P\left(\xi^2\right).
\tag{31}
$$

If for either perturbation $\xi^1$ or $\xi^2$ the problem becomes infeasible then $P(\xi^1) = -\infty$ or $P(\xi^2) = -\infty$ and thus Equation 31 holds trivially. For perturbations which keep the problem feasible, suppose $P(\xi^1)$ and $P(\xi^2)$ are achieved by the policies $\mu_1 \in \mathcal{P}(\mathcal{S})$ and $\mu_2 \in \mathcal{P}(\mathcal{S})$ respectively. Then,

$P(\xi^1) = \mathcal{J}(\mu_1)$ with $-\bar{H}(\pi_b, \mu_1) + \kappa_0 \geq \xi^1$ and $P(\xi^2) = \mathcal{J}(\mu_2)$ with $-\bar{H}(\pi_b, \mu_2) + \kappa_0 \geq \xi^2$. To establish Equation 31 it suffices to show that for every $t \in (0, 1)$ there exists a policy $\mu_t$ such that $-\bar{H}(\pi_b, \mu_t) + \kappa_0 \geq t\xi^1 + (1 - t)\xi^2$ and $\mathcal{J}(\mu_t) = t\mathcal{J}(\mu_1) + (1 - t)\mathcal{J}(\mu_2)$. Notice that any policy $\mu_t$ satisfying the previous conditions is a feasible policy for the slack $-\kappa_0 + t\xi^1 + (1 - t)\xi^2$. Hence, by definition of the perturbed function Equation 31, it follows that

$$P\left[t\xi^1 + (1 - t)\xi^2\right] \geq \mathcal{J}(\mu_t) = t\mathcal{J}(\mu_1) + (1 - t)\mathcal{J}(\mu_2) = tP\left(\xi^1\right) + (1 - t)P\left(\xi^2\right). \quad (32)$$

If such a policy exists, the previous equation implies Equation 31. Thus, to complete the proof of the result we need to establish its existence. To do so we start by formulating the objective $\mathcal{J}(\mu)$ as a linear function.

After sufficient training, we can suppose $Q_\phi(\boldsymbol{s}, \boldsymbol{a})$ as an unbiased approximation of $Q^\mu(\boldsymbol{s}, \boldsymbol{a})$,

$$Q_\phi(\boldsymbol{s}, \boldsymbol{a}) \approx Q^\mu(\boldsymbol{s}, \boldsymbol{a}) = \mathbb{E}_{\tau \sim d_0, \mu, \mathcal{T}}\left[\sum_{j=0}^{J} \gamma^j r(\boldsymbol{s}_j, \boldsymbol{a}_j | \boldsymbol{s}_0 = \boldsymbol{s}, \boldsymbol{a}_0 = \boldsymbol{a})\right], \quad (33)$$

where $\tau$ denotes the trajectory generated by policy $\mu(\boldsymbol{a}|\boldsymbol{s})$, start state distribution $d_0$ and environment dynamics $\mathcal{T}(\boldsymbol{s}'|\boldsymbol{s}, \boldsymbol{a})$.

So the objective function Equation 6 can be written as

$$
\begin{aligned}
J(\mu) &= \int_{\mathcal{S}} d_0(\boldsymbol{s}) \int_{\mathcal{A}} Q^\mu(\boldsymbol{s}, \boldsymbol{a}) d\boldsymbol{a} d\boldsymbol{s} \\
&= \sum_{t=0}^{\infty} \gamma^t \mathbb{E}_{\boldsymbol{s}_t \sim p_\mu(\boldsymbol{s}_t = s)}\left[\mathbb{E}_{\boldsymbol{a} \sim \mu(\boldsymbol{a}_t|\boldsymbol{s}_t)} r(\boldsymbol{s}_t, \boldsymbol{a}_t)\right] \\
&= \sum_{t=0}^{\infty} \gamma^t \left[\int_{\mathcal{S} \times \mathcal{A}} p_\mu(\boldsymbol{s}_t = \boldsymbol{s}) \mu(\boldsymbol{a}_t = \boldsymbol{a}|\boldsymbol{s}_t = \boldsymbol{s}) r(\boldsymbol{s}, \boldsymbol{a}) d\boldsymbol{a} d\boldsymbol{s}\right] \\
&= \int_{\mathcal{S} \times \mathcal{A}} \left[\sum_{t=0}^{\infty} \gamma^t p_\mu(\boldsymbol{s}_t = \boldsymbol{s}) \mu(\boldsymbol{a}_t = \boldsymbol{a}|\boldsymbol{s}_t = \boldsymbol{s}) r(\boldsymbol{s}, \boldsymbol{a}) d\boldsymbol{a} d\boldsymbol{s}\right] \\
&= \frac{1}{1 - \gamma} \mathbb{E}_{\boldsymbol{s}, \boldsymbol{a} \sim \rho^\mu(\boldsymbol{s}, \boldsymbol{a})}\left[r(\boldsymbol{s}, \boldsymbol{a})\right] \\
&= \frac{1}{1 - \gamma} \int_{\mathcal{S} \times \mathcal{A}} \rho^\mu(\boldsymbol{s}, \boldsymbol{a}) r(\boldsymbol{s}, \boldsymbol{a}) d\boldsymbol{a} d\boldsymbol{s},
\end{aligned}
\quad (34)
$$

which implies the objective function Equation 6 equivalent to

$$\max_{\rho \in \mathcal{R}} \frac{1}{1 - \gamma} \int_{\mathcal{S} \times \mathcal{A}} \rho^\mu(\boldsymbol{s}, \boldsymbol{a}) r(\boldsymbol{s}, \boldsymbol{a}) d\boldsymbol{a} d\boldsymbol{s}, \quad (35)$$

where $\mathcal{R}$ is the set of occupation measures, which is convex and compact.[ Borkar (1988), Theorem 3.1]

So Equation 35 is a linear function on $\rho(\boldsymbol{s}, \boldsymbol{a})$. Let $\rho_1(\boldsymbol{s}, \boldsymbol{a}), \rho_2(\boldsymbol{s}, \boldsymbol{a}) \in \mathcal{R}$ be the occupation measures associated to $\mu_1$ and $\mu_2$. Since, $\mathcal{R}$ is convex, there exists a policy $\mu_t$ such that its corresponding occupation measure is $\rho_t(\boldsymbol{s}, \boldsymbol{a}) = t\rho_1(\boldsymbol{s}, \boldsymbol{a}) + (1 - t)\rho_2(\boldsymbol{s}, \boldsymbol{a}) \in \mathcal{R}$ and $\mathcal{J}(\mu_t) = t\mathcal{J}(\mu_1) + (1 - t)\mathcal{J}(\mu_2)$. To prove Equation 31, it suffices to show such $\mu_t$ satisfies $-\bar{H}(\pi_b, \mu) + \kappa_0 \geq t\xi^1 + (1 - t)\xi^2$.

$$-\bar{H}(\pi_b, \mu) = \frac{1}{1-\gamma} \int_{\mathcal{S}\times\mathcal{A}} \rho^{\pi_b}(\boldsymbol{s}, \boldsymbol{a}) \log \frac{\rho_t^{\mu}(\boldsymbol{s}, \boldsymbol{a})}{\rho^{\pi_b}(\boldsymbol{s})} d\boldsymbol{a}d\boldsymbol{s}$$

$$= \frac{1}{1-\gamma} \int_{\mathcal{S}\times\mathcal{A}} \rho^{\pi_b}(\boldsymbol{s}, \boldsymbol{a}) \log \frac{t\rho_1(\boldsymbol{s}, \boldsymbol{a}) + (1-t)\rho_2(\boldsymbol{s}, \boldsymbol{a})}{\rho^{\pi_b}(\boldsymbol{s})} d\boldsymbol{a}d\boldsymbol{s}$$

$$\geq \underbrace{\frac{1}{1-\gamma} t \int_{\mathcal{S}\times\mathcal{A}} \rho^{\pi_b}(\boldsymbol{s}, \boldsymbol{a}) \log \frac{\rho_1(\boldsymbol{s}, \boldsymbol{a})}{\rho^{\pi_b}(\boldsymbol{s})} d\boldsymbol{a}d\boldsymbol{s} + }_{\text{Jensen's inequality.}}$$

$$\underbrace{\frac{1}{1-\gamma}(1-t) \int_{\mathcal{S}\times\mathcal{A}} \rho^{\pi_b}(\boldsymbol{s}, \boldsymbol{a}) \log \frac{\rho_2(\boldsymbol{s}, \boldsymbol{a})}{\rho^{\pi_b}(\boldsymbol{s})} d\boldsymbol{a}d\boldsymbol{s}}_{\text{Jensen's inequality.}} \quad (36)$$

$$\geq t\xi^1 + (1-t)\xi^2 - \kappa_0$$

This completes the proof that the perturbation function is concave and according to Lemma A.1 strong duality for Equation 12 holds.

Next, we show that Slater's condition in Theorem 3.2 is a mild condition, since for all $\kappa_0 > -E_{\boldsymbol{s}\sim\rho^{\pi_b}(\boldsymbol{s}), \boldsymbol{a}\sim\pi_b(\cdot|\boldsymbol{s})} [\log \pi_b(\cdot|\boldsymbol{s})]$, we can get $\mu = \pi_b$ is strictly feasible, which meets Slater's conditions.

Finally, we prove Equation 12, which has the same optimal policy $\mu$ with Equation 14.

For Equation 14, if strong duality holds and a dual optimal solution $\lambda^*$ exists, then any primal optimal point of Equation 12 is also a maximizer of $\mathcal{L}(\mu, \lambda^*)$, this is obvious through the KKT conditions.[ Boyd & Vandenberghe (2004) Ch.5.5.5.]

So if we can prove that the solution of the $\mathcal{L}(\mu, \lambda^*)$ is unique, we can compute the optimal policy of Equation 12 from a dual problem Equation 14:

$$\mathcal{L}(\mu, \lambda^*) = J(\mu) + \lambda^*(\kappa_0 - \bar{H}(\pi_b, \mu))$$

$$= \frac{1}{1-\gamma}\left[\underbrace{\int_{\mathcal{S}\times\mathcal{A}} \rho^{\mu}(\boldsymbol{s}, \boldsymbol{a})r(\boldsymbol{s}, \boldsymbol{a})d\boldsymbol{a}d\boldsymbol{s}}_{\text{Using Equation 34.}} + \lambda^*(\bar{\kappa_0} - \int_{\mathcal{S}\times\mathcal{A}} \rho^{\pi_b}(\boldsymbol{s}, \boldsymbol{a}) \log \frac{\rho^{\mu}(\boldsymbol{s}, \boldsymbol{a})}{\rho^{\pi_b}(\boldsymbol{s})}d\boldsymbol{a}d\boldsymbol{s})\right].$$

$$(37)$$

According to Equation 34 $J(\mu)$ is a linear function, and negative relative entropy is concave, this implies that $\mathcal{L}(\mu, \lambda^*)$ is a strictly concave function of $\rho^{\mu}$ with only a unique maximum value and since we have that the occupancy measure $\rho^{\mu}(\boldsymbol{s}, \boldsymbol{a})$ has a one-to-one relationship with policy $\mu$, we can obtain the unique $\mu$ corresponding to $\rho^{\mu}$. $\qquad\square$

## A.3 PROPOSITION 3.1

Finally, we prove the Proposition 3.1.

**Proof.**

$$H(\pi_b, \mu) = -\mathbb{E}_{\boldsymbol{s}\sim\rho^{\pi_b}(\boldsymbol{s}), \boldsymbol{a}\sim\pi_b(\cdot|\boldsymbol{s})} [\log \mu(\cdot|\boldsymbol{s})]$$

$$= -\mathbb{E}_{\boldsymbol{s}\sim\rho^{\pi_b}(\boldsymbol{s})}\left[\int \pi_b(\boldsymbol{a}|\boldsymbol{s})d\boldsymbol{a} \log \int \pi_b(\boldsymbol{a}^{1:T}|\boldsymbol{a}, \boldsymbol{s})\frac{\mu(\boldsymbol{a}^{0:T}|\boldsymbol{s})}{\pi_b(\boldsymbol{a}^{1:T}|\boldsymbol{a}, \boldsymbol{s})}d\boldsymbol{a}^{1:T}\right]$$

$$\leq \mathbb{E}_{\boldsymbol{s}\sim\rho^{\pi_b}(\boldsymbol{s})}\left[\underbrace{\int \pi_b(\boldsymbol{a}^{0:T}) \log[\frac{\pi_b(\boldsymbol{a}^{1:T}|\boldsymbol{a}, \boldsymbol{s})}{\mu(\boldsymbol{a}^{0:T}|\boldsymbol{s})}]d\boldsymbol{a}^{0:T}}_{\text{Jensen's inequality}}\right] = K. \quad (38)$$

$H(\pi_b, \mu) = K$ is true if and only if $\pi_b(\boldsymbol{a}^{1:T}|\boldsymbol{a}^0, \boldsymbol{s}) = \mu(\boldsymbol{a}^{1:T}|\boldsymbol{a}^0, \boldsymbol{s})$. By letting the KL divergence between $\pi_b$ and $\mu$ be small enough, we can approximate the $H(\pi_b, \mu)$ with $K$. Furthermore, we can use the variational lower bound $K$ to approximate the KL divergence (Ho et al., 2020):

$$H(\pi_b, \mu) \approx K = \mathbb{E}_{\boldsymbol{s} \sim \rho^{\pi_b}(\boldsymbol{s})} \left[ \mathbb{E}_{\pi_b(\boldsymbol{a}^{0:T})} \left[ \log \frac{\pi_b(\boldsymbol{a}^{1:T}|\boldsymbol{a}, \boldsymbol{s})}{\mu(\boldsymbol{a}^{0:T}|\boldsymbol{s})} \right] \right] = \mathbb{E}_{\boldsymbol{s} \sim \rho^{\pi_b}(\boldsymbol{s})} \left[ \mathbb{E}_{\pi_b(\boldsymbol{a}^{0:T})} \left[ \log \frac{\pi_b(\boldsymbol{a}^{1:T}|\boldsymbol{a}^0, \boldsymbol{s})}{\mu(\boldsymbol{a}^{0:T}|\boldsymbol{s})} \right] \right]$$

$$= \mathbb{E}_{\rho^{\pi_b}(\boldsymbol{s}, \boldsymbol{a})} \left[ \log \frac{\prod_{t=1}^{T} \pi_b(\boldsymbol{a}^t|\boldsymbol{a}^{t-1})}{\mu(\boldsymbol{a}^T) \prod_{t=1}^{T} \mu(\boldsymbol{a}^{t-1}|\boldsymbol{a}^t)} \right]$$

$$= \mathbb{E}_{\rho^{\pi_b}(\boldsymbol{s}, \boldsymbol{a})} \left[ -\log \mu(\boldsymbol{a}^T) + \sum_{t=1}^{T} \log \frac{\pi_b(\boldsymbol{a}^t|\boldsymbol{a}^{t-1})}{\mu(\boldsymbol{a}^{t-1}|\boldsymbol{a}^t)} \right]$$

$$= \mathbb{E}_{\rho^{\pi_b}(\boldsymbol{s}, \boldsymbol{a})} \left[ -\log \mu(\boldsymbol{a}^T) + \sum_{t=2}^{T} \log \frac{\pi_b(\boldsymbol{a}^t|\boldsymbol{a}^{t-1})}{\mu(\boldsymbol{a}^{t-1}|\boldsymbol{a}^t)} + \log \frac{\pi_b(\boldsymbol{a}^1|\boldsymbol{a}^0)}{\mu(\boldsymbol{a}^0|\boldsymbol{a}^1)} \right]$$

$$= \mathbb{E}_{\rho^{\pi_b}(\boldsymbol{s}, \boldsymbol{a})} \left[ -\log \mu(\boldsymbol{a}^T) + \sum_{t=2}^{T} \log \left( \frac{\pi_b(\boldsymbol{a}^{t-1}|\boldsymbol{a}^t, \boldsymbol{a}^0)}{\mu(\boldsymbol{a}^{t-1}|\boldsymbol{a}^t)} \cdot \frac{\pi_b(\boldsymbol{a}^t|\boldsymbol{a}^0)}{\pi_b(\boldsymbol{a}^{t-1}|\boldsymbol{a}^0)} \right) + \log \frac{\pi_b(\boldsymbol{a}^1|\boldsymbol{a}^0)}{\mu(\boldsymbol{a}^0|\boldsymbol{a}^1)} \right]$$

$$= \mathbb{E}_{\rho^{\pi_b}(\boldsymbol{s}, \boldsymbol{a})} \left[ -\log \mu(\boldsymbol{a}^T) + \sum_{t=2}^{T} \log \frac{\pi_b(\boldsymbol{a}^{t-1}|\boldsymbol{a}^t, \boldsymbol{a}^0)}{\mu(\boldsymbol{a}^{t-1}|\boldsymbol{a}^t)} + \sum_{t=2}^{T} \log \frac{\pi_b(\boldsymbol{a}^t|\boldsymbol{a}^0)}{\pi_b(\boldsymbol{a}^{t-1}|\boldsymbol{a}^0)} + \log \frac{\pi_b(\boldsymbol{a}^1|\boldsymbol{a}^0)}{\mu(\boldsymbol{a}^0|\boldsymbol{a}^1)} \right]$$

$$= \mathbb{E}_{\rho^{\pi_b}(\boldsymbol{s}, \boldsymbol{a})} \left[ -\log \mu(\boldsymbol{a}^T) + \sum_{t=2}^{T} \log \frac{\pi_b(\boldsymbol{a}^{t-1}|\boldsymbol{a}^t, \boldsymbol{a}^0)}{\mu(\boldsymbol{a}^{t-1}|\boldsymbol{a}^t)} + \log \frac{\pi_b(\boldsymbol{a}^T|\boldsymbol{a}^0)}{\pi_b(\boldsymbol{a}^1|\boldsymbol{a}^0)} + \log \frac{\pi_b(\boldsymbol{a}^1|\boldsymbol{a}^0)}{\mu(\boldsymbol{a}^0|\boldsymbol{a}^1)} \right]$$

$$= \mathbb{E}_{\rho^{\pi_b}(\boldsymbol{s}, \boldsymbol{a})} \left[ \log \frac{\pi_b(\boldsymbol{a}^T|\boldsymbol{a}^0)}{\mu(\boldsymbol{a}^T)} + \sum_{t=2}^{T} \log \frac{\pi_b(\boldsymbol{a}^{t-1}|\boldsymbol{a}^t, \boldsymbol{a}^0)}{\mu(\boldsymbol{a}^{t-1}|\boldsymbol{a}^t)} - \log \mu(\boldsymbol{a}^0|\boldsymbol{a}^1) \right]$$

$$= \mathbb{E}_{\rho^{\pi_b}(\boldsymbol{s}, \boldsymbol{a})} [\underbrace{D_{\mathrm{KL}}(\pi_b(\boldsymbol{a}^T|\boldsymbol{a}^0) \,\|\, \mu(\boldsymbol{a}^T))}_{L_T} + \sum_{t=2}^{T} \underbrace{D_{\mathrm{KL}}(\pi_b(\boldsymbol{a}^{t-1}|\boldsymbol{a}^t, \boldsymbol{a}^0) \,\|\, \mu(\boldsymbol{a}^{t-1}|\boldsymbol{a}^t))}_{L_{t-1}} \underbrace{-\log \mu(\boldsymbol{a}^0|\boldsymbol{a}^1)}_{L_0}].$$

So we can approximate the entropy constraint in Equation 12:

$$H(\pi_b, \mu) \approx K = L_T + \sum_{t=2}^{T} L_{t-1} + L_0. \tag{39}$$

Following DDPM (Ho et al., 2020) use the random sample to approximate the $\sum_{t=2}^{T} L_{t-1} + L_0$, we have

$$\sum_{t=2}^{T} L_{t-1} + L_0 \approx \mathcal{L}_c(\pi_b, \mu)$$
$$= \mathbb{E}_{i \sim [1:N], \epsilon \sim \mathcal{N}(\boldsymbol{0}, \boldsymbol{I}), (\boldsymbol{s}, \boldsymbol{a}) \sim \mathcal{D}} \left[ ||\epsilon - \epsilon_\theta(\sqrt{\bar{\alpha}_i}\boldsymbol{a} + \sqrt{1-\bar{\alpha}_i}\epsilon, \boldsymbol{s}, i)||^2 \right]. \tag{40}$$

Let $L_T = c$, combining Equation 39 and Equation 40 we can get

$$H(\pi_b, \mu) \approx c + \mathcal{L}_c(\pi_b, \mu). \tag{41}$$

$\square$

# B  NOTATION TABLE

# C  EXPERIMENTAL DETAILS

We follow the Wang et al. (2022) to build our policy with an MLP-based conditional diffusion model. Following the DDPM, we recover the action from a residual network $\epsilon(\boldsymbol{a}^i, \boldsymbol{s}, i)$, and we model the $\epsilon_\theta$ as a 3-layers MLPs with Mish activations. All the hidden units have been set to 256. We also follow the Fujimoto et al. (2018) to build two Q networks with the same MLP setting as our critic network. All the networks are optimized through Adam (Kingma & Ba, 2014). We provide all the hyperparameters in Table 4.

$$
\begin{aligned}
H(\pi_b, \mu) &\triangleq -\mathbb{E}_{\boldsymbol{s}\sim\rho^{\pi_b}(\boldsymbol{s}),\boldsymbol{a}\sim\pi_b(\cdot|\boldsymbol{s})}\left[\log\mu(\cdot|\boldsymbol{s})\right].\\
\mu, \mu(\boldsymbol{a}|\boldsymbol{s}) \text{ or } \mu(\cdot|\boldsymbol{s}) &\triangleq \text{policy.}\\
\pi_b, \pi_b(\boldsymbol{a}|\boldsymbol{s}) \text{ or } \pi_b(\cdot|\boldsymbol{s}) &\triangleq \text{behavior policy.}\\
p_\mu(\boldsymbol{s}_t = \boldsymbol{s}) &\triangleq \text{Probability of landing in state } \boldsymbol{s} \text{ at time } t, \text{ when following policy } \mu\\
&\quad \text{from an initial state sampled from } d_0, \text{ in an environment with transition}\\
&\quad \text{dynamics } \mathcal{T}.\\
\mathcal{T}, \mathcal{P}(\boldsymbol{s}'|\boldsymbol{s},\boldsymbol{a}) \text{ or } p(\boldsymbol{s}'|\boldsymbol{s},\boldsymbol{a}) &\triangleq \text{transition dynamics.}\\
d_0(\boldsymbol{s}) &\triangleq \text{initial state distribution of behavior policy.}\\
\rho^\mu(\boldsymbol{s}) &\triangleq \rho^\mu(\boldsymbol{s}) = \sum_{t=0}^{\infty}\gamma^t p_\mu(\boldsymbol{s}_t = \boldsymbol{s}) \text{ is the unnormalized discounted state vis-}\\
&\quad \text{itation frequencies.}\\
\rho^\mu(\boldsymbol{s},\boldsymbol{a}) &\triangleq \rho^\mu(\boldsymbol{s},\boldsymbol{a}) = (1-\gamma)\sum_{t=0}^{\infty}\gamma^t p_\mu(\boldsymbol{s}_t = \boldsymbol{s})\mu(\boldsymbol{a}|\boldsymbol{s}) = (1-\gamma)\rho^\mu(\boldsymbol{s})\mu(\boldsymbol{a}|\boldsymbol{s})\\
&\quad \text{is the occupation measure of policy } \mu.\\
Q_\phi(\boldsymbol{s},\boldsymbol{a}) &\triangleq \text{parameterized state action function.}\\
Q_{\phi'}(\boldsymbol{s},\boldsymbol{a}) &\triangleq \text{parameterized target state action function.}\\
\mu_\theta(\boldsymbol{a}|\boldsymbol{s}) &\triangleq \text{parameterized policy.}\\
\mu_{\theta'}(\boldsymbol{a}|\boldsymbol{s}) &\triangleq \text{parameterized target policy.}\\
\epsilon_\theta(\boldsymbol{x}_i, i) &\triangleq \text{parameterized Gaussian noise predict network in diffusion model.}\\
f_\phi(\boldsymbol{y}|\boldsymbol{x}_i) &\triangleq \text{parameterized classifier.}
\end{aligned}
$$

Table 3: Table of Notation in paper

| Dataset | Environment | Learning Rate | $\kappa$ | Reward Tune | max Q backup | Policy evaluation interval | $\lambda_{\text{clip}}$ |
|---|---|---|---|---|---|---|---|
| Medium-Expert | HalfCheetah | 3e-4 | 0.04 | None | False | 2 | 0 |
| Medium-Expert | Hopper | 3e-4 | 0.03 | None | False | 2 | 0 |
| Medium-Expert | Walker2d | 3e-4 | 0.04 | None | False | 2 | 0 |
| Medium | HalfCheetah | 3e-4 | 0.06 | None | False | 2 | 0 |
| Medium | Hopper | 3e-4 | 0.05 | None | False | 2 | 0 |
| Medium | Walker2d | 3e-4 | 0.03 | None | False | 2 | 0 |
| Medium-Replay | HalfCheetah | 3e-4 | 0.06 | None | False | 2 | 0 |
| Medium-Replay | Hopper | 3e-4 | 0.03 | None | False | 2 | 0 |
| Medium-Replay | Walker2d | 3e-4 | 0.03 | None | False | 2 | 0 |
| Default | AntMaze-umaze | 3e-4 | 0.2 | CQL | False | 2 | 0.3 |
| Diverse | AntMaze-umaze | 3e-4 | 0.09 | CQL | True | 2 | 0.3 |
| Play | AntMaze-medium | 1e-3 | 0.3 | CQL | True | 2 | 0.3 |
| Diverse | AntMaze-medium | 3e-4 | 0.2 | CQL | True | 2 | 0.3 |
| Play | AntMaze-large | 3e-4 | 0.2 | CQL | True | 4 | 0.3 |
| Diverse | AntMaze-large | 3e-4 | 0.2 | CQL | True | 4 | 0.3 |

Table 4: Hyperparameter settings of all selected tasks. Reward Tune with CQL means that we use the reward tuning method in Kumar et al. (2020). We also use max Q backup to improve performance in AntMaze tasks. In offline reinforcement learning, it's common to fine-tune rewards for the AntMaze tasks. Additionally, we have observed that some other diffusion-based methods, such as SfBC, perform more reward engineering compared to our method. $\lambda_{\text{clip}}$ in our paper means that $\lambda \geq \lambda_{\text{clip}}$.

We train for 1000 epochs (2000 for Gym tasks). Each epoch consists of 1000 gradient steps with batch size 256. The training in MuJoCo locomotion is usually quite stable as shown in Figure 2. However, the training for AntMaze is relatively unstable due to its sparse reward setting and suboptimal trajectories in the offline datasets.

**Runtime.** We test the runtime of DiffCPS on a RTX 3050 GPU. For algorithm training, the runtime cost of training the gym Locomotion tasks is about 4h26min for 2000 epochs (2e6 gradient steps), see Table 5 for details.

| D4RL Tasks | DiffCPS (T=5) | DiffusionQL (T=5) | SfBC (T=5) |
|---|---|---|---|
| **Locomotion Runtime** (1 epoch) | 7.68s | 5.1s | 8.42s |
| **AntMaze Runtime** (1 epoch) | 9.96s | 10.5s | 10.53s |

Table 5: Runtime of different diffusion-based offline RL methods.

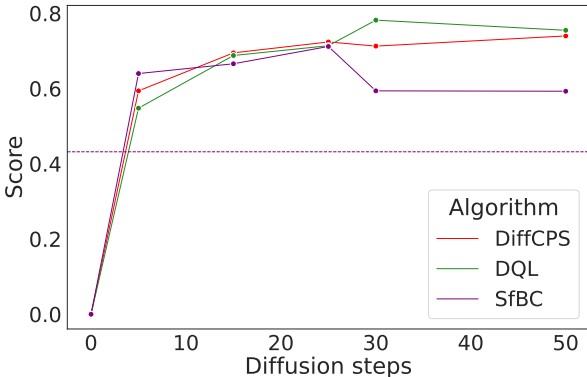

Figure 5: Evaluation performance of DiffCPS and other baselines on toy bandit experiments. The dashed line represents the score of AWR. We also observe that as $T$ increases, diffusion-based algorithms all experience a certain degree of performance decline, especially SfBC. The reason could be that as $T$ increases, the increased model capacity leads to overfitting the data in the dataset. In the case of SfBC, the presence of sampling errors exacerbates this phenomenon.

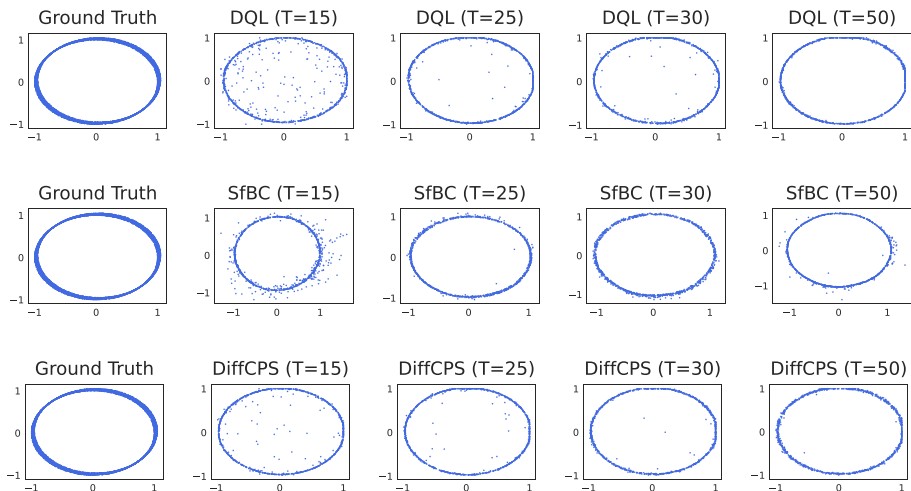

Figure 6: Evaluation effect of different diffusion steps on diffusion-based algorithms.

From Table 5, it's evident that the runtime of DiffCPS is comparable to other diffusion model-based methods. Further optimization like using Jax and incorporating other diffusion model acceleration tricks can improve the runtime of DiffCPS.

# D  MORE TOY EXPERIMENTS

In this chapter, we provide further detailed information about the toy experiments. The noisy circle is composed of a unit circle with added Gaussian noise $\mathcal{N}(0, 0.05)$. The offline data $\mathcal{D} = \{(\boldsymbol{a}_i)\}_{i=1}^{5000}$ are collected from the noisy circle. We train all methods with 20,000 steps to ensure convergence. The network framework of SfBC remains consistent with the original SfBC paper. Both DQL and DiffCPS use the same MLP architecture. The code for DQL and SfBC is sourced from the official code provided by the authors, while the code for AWR is adopted from a PyTorch implementation available on GitHub.

In Figure 6, we put the full results of diffusion-based methods and Figure 5 displays the scores of diffusion-based algorithms under different diffusion steps, denoted as $T$. The score calculation involves a modified Jaccard Index, which computes how many actions fall into the offline dataset (the minimum distance is less than 1e-8). The dashed line represents the score of AWR.

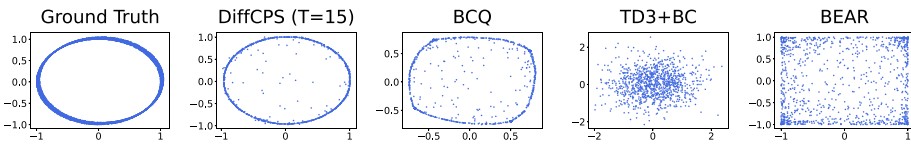

Figure 7: Evaluation effect of prior offline methods.

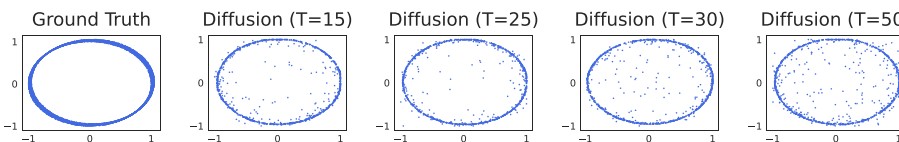

Figure 8: Results of diffusion behavior clone.

In Figure 7, we compare DiffCPS with prior offline RL methods, including the TD3+BC (Fujimoto & Gu, 2021) with Gaussian policy, BCQ (Fujimoto et al., 2019), and BEAR (Kumar et al., 2019). Note that both BCQ and BEAR use the VAE to model behavior policy, and then get the policy by regularizing the cloned behavior policy. However, results in Figure 7 show that VAE cannot exactly model the multi-modal data in offline datasets. TD3+BC with Gaussian policy also fails to model the optimal behavior due to the limited policy expressivity described in Section 3.1.

In Figure 8, we show the results of diffusion-based behavior clone. The results in Figure 8 and Figure 7 show that the diffusion model can model multi-modal distribution while other methods struggle to capture the multi-modal behavior policy.

## E  EXTRA RELATED WORK

**Offline Model-based RL.** Model-based RL methods represent another approach to address offline reinforcement learning. Similar to model-free offline RL, model-based RL also needs to address extrapolation error. Kidambi et al. (2020); Yu et al. (2020) perform pessimistic planning in the learned model, which penalizes the reward for uncertainty. BREMEN (Matsushima et al.) uses trust region constraint to update policy in an ensemble of dynamic models to avoid the extrapolation error. VL-LCB (Rashidinejad et al., 2021) employs offline value iteration with lower confidence bound to address the extrapolation error. SGP (Suh et al., 2023) uses the score function of the diffusion model to approximate the gradient of the proposed uncertainty estimation. Our method differs from them for DiffCPS tackles the diffusion-based constrained policy search problem in offline RL through Lagrange dual and recursive optimization.

## F  EXTRA EXPERIMENTS

### F.1  TRAJECTORY STITCHING

The offline dataset is depicted in the Figure 9. The reward function is the sum of the negative Euclidean distance from the point $(1, 1)$ and a constant, with both actions and states represented as 2D coordinates. The dataset consists only of trajectories forming an X shape. During the evaluation, the starting point is at $(-1, 1)$. To achieve maximum reward, the policy needs to stitch together offline data to reach the goal. The trajectories generated by our policy are shown in Figure 10. We note that DiffCPS successfully combines trajectories from the dataset to reach the vicinity of the target point $(1, 1)$ while avoiding low-reward areas.

### F.2  TRIFINGER DATASET EXPERIMENT

The TriFinger dataset (Gürtler et al.) is an offline dataset for the TriFinger robotic arm that includes both real and simulated data. The dataset primarily focuses on two tasks: Push and Lift. We utilize the Push-Sim-Expert dataset to train our DiffCPS, and the network architecture is consistent with the details provided in Appendix C. The goal of the push task is to move the cube to a target location.

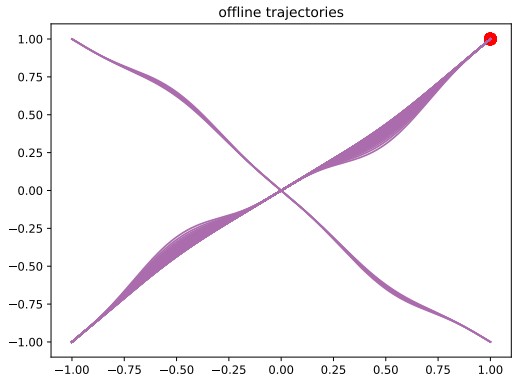

Figure 9: Offline Dataset.

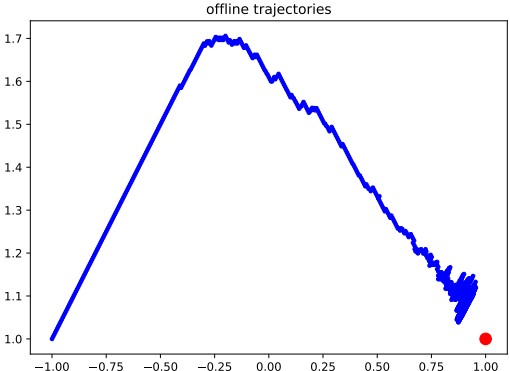

Figure 10: Trajectory of DiffCPS.

This task does not require the agent to align the orientation of the cube; the reward is based only on the desired and the achieved position.

When trained with 2e6 training steps (gradient steps) on the sim-expert dataset, DiffCPS ($T = 45$, target_kl = 0.01, $\lambda_{\min} = 0$) achieves a success rate of $0.906 \pm 0.001$ in the push task based on 100 evaluation episodes and 5 random seeds, as detailed in the Table 6.

| TriFinger-Push | BC | CRR | IQL | DiffCPS (Ours) |
|---|---|---|---|---|
| Sim-Expert | $0.83 \pm 0.02$ | $\mathbf{0.94 \pm 0.04}$ | $0.88 \pm 0.04$ | $\mathbf{0.906 \pm 0.001}$ |

Table 6: The success rate of DiffCPS and other baselines on TriFinger Push task. Results for other baselines are sourced from Gürtler et al. and have been carefully tuned. It's worth noting that our DiffCPS still utilizes a simple MLP to represent the policy, suggesting potential benefits from more powerful network architectures.

## G AUGMENTED LAGRANGIAN METHOD

In Algorithm 1, we utilize alternating optimization (dual ascent) to solve the dual problem in Equation 14. In practice, we can employ the Augmented Lagrangian Method (ALM) to enhance the stability of the algorithm. ALM introduces a strongly convex term (penalty function) into the Lagrangian function to improve convergence. Our experimental results suggest that ALM can enhance training stability but may lead to a performance decrease. This could be attributed to (i) the introduction of the penalty function altering the optimal KL divergence constraint value and (ii) the penalty function making the policy too conservative, thereby preventing the adoption of high-reward actions. We believe that further research will benefit our algorithm by incorporating ALM and other optimization algorithms.

