# OpenReview forum: "DiffCPS: Diffusion Model based Constrained Policy Search for Offline Reinforcement Learning"
_ICLR.cc/2024/Conference — Submitted to ICLR 2024_

### Official Review · Reviewer_4x5N · 2023-10-17

**Soundness:** 2 fair
**Presentation:** 2 fair
**Contribution:** 2 fair
**Rating:** 3
**Confidence:** 5

**Summary:**

The authors propose a new algorithm called DiffCPS for solving offline RL problems. They claim that DiffCPS can learn a diffusion-based policy avoiding the difficult density calculation brought by traditional AWR framework. They present theoretical justification for their approach and perform an empirical study to validate their proposed algorithms.

**Strengths:**

* The numerical experiments are well designed, with a detailed section of ablation study. Also, the toy example is intuitive and interesting.

**Weaknesses:**

* Corollary 3.1.1 and Theorem 3.2 are flawed. In the proof, the authors claim $J(\mu)$ is an affine function of $\mu$, which is **false**. In fact, $J(\mu)$ is **not** convex w.r.t. $\mu$ and the duality results can not hold. Consequently, the theoretical analysis presented in this paper is compromised, significantly undermining its contributions.

* The paper lacks clarity. Multiple mathematical objects are introduced without clear definitions. For example, $Q_\phi(s,a)$, $\epsilon_\theta(x_i,i)$, $f_\phi(y\mid x_i)$, $H(\cdot,\cdot)$, etc. . If the reader is not familiar with the literature on RL and diffusion models, she/he will definitely get confused by the undefined notations. Also, I think the authors should provide more explanations about the exact methods used to update $\mu$ and $\lambda$. It would be appreciated if the authors could give a complete and precise description of DiffCPSS. (If there are problems with page limits the authors can put it in the appendix.)

* The empirical performance of the proposed algorithm exhibits only marginal improvement when compared to the baselines.

**Questions:**

* The authors deploy diffusion-based policies in place of the traditional Gaussian policies to handle the multi-modal problem. I wonder whether there are choices other than diffusion models and naive Gaussians. For example, flow models can express a rich distribution class and have explicit densities. Is it possible to fit the flow models into the framework of AWR?

* In the toy example, the authors claim that "SfBC incorrectly models the dataset as a circle instead of a noisy circle". I checked the image and did not find such a difference. Can the authors give an explanation?

---

> ### Author Response · Authors · 2023-11-21
> **Response to 4X5N**
>
> We thank reviewer 4X5N for pointing out the drawbacks of our paper. Below, we address the concerns raised in your review point by point. Please let us know if you have any further concerns or whether this adequately addresses all the issues that you raised with the paper.
> >   **Q1:** Corollary 3.1.1 and Theorem 3.2 are flawed.
>
>   **A1:** Thanks for pointing this out.  We have rigorously re-proven the strong duality of Equation 12 in Appendix A.2. New proof relies on a well-known result from perturbation theory connecting strong duality to the convexity of the perturbation function. Here, we formally state the lemma used in the proof,  which is referred to [1] Cor. 30.2.2 and [2] Ch 3.
> >If (i) $r$ is bounded; (ii) Slater's condition holds for the diffusion-based constrained policy search and (iii) its perturbation function~$P(\xi)$ is concave, then strong duality holds for equation 12.
>
> The proof consists of two main steps (See details in Appendix A.2. in our revision): (i) demonstrating that the perturbation problem of the simplified diffusion-based constrained search problem, as per Theorem 3.1, is a concave function, and (ii) proving the Slater's condition holds for the diffusion-based constrained search problem. We have also rephrased the main theorems of the paper to ensure a more coherent logical flow.
>
> >   **Q2:** The paper lacks clarity. Multiple mathematical objects are introduced without clear definitions. If the reader is not familiar with the literature on RL and diffusion models, she/he will definitely get confused by the undefined notations. Also, I think the authors should provide more explanations about the exact methods used to update $\mu$ and $\lambda$ . It would be appreciated if the authors could give a complete and precise description of DiffCPSS. (If there are problems with page limits the authors can put it in the Appendix.)
>
>   **A2:** Thanks for suggesting this. We have added the Notation table in Appendix B and rephrased our DiffCPS algorithm in Chapter 3. Here, we reiterate the logic of our algorithm. We first establish the strong duality for Equation 12, allowing us to use Equation 14 to find the optimal policy. The approach to solving Equation 14 follows SAC [3], using alternating optimization to optimize the policy and Lagrange multipliers separately. Equation 19 and Equation 20 represent the loss functions for the Lagrange multipliers and the policy, respectively.
>
> >  **Q3:**  The empirical performance of the proposed algorithm exhibits only marginal improvement when compared to the baselines.
>
>
>
> **A3:**
> | D4RL Tasks              | DiffCPS (Ours)  | DiffusionQL [6] (ICLR 2023) | SfBC [4] (ICLR 2023)   |IQL [5] (ICLR 2022)|
> |-------------------------|---------------|-------------------|-------------|-----------|
> | Locomotion Average      | 92.26         | 87.9            | 75.6       |76.9|
> | AntMaze Average       | 81.69         | 69.8             | 74.2      |63.0|
>
> **Table 1:** D4RL performance of different offline RL methods.
> The table above illustrates the performance of recent ICLR offline RL algorithms. SfBC (ICLR 2023) shows a similar performance to IQL (ICLR 2022), with an average decrease of 1.7% in locomotion tasks and a 17.8% improvement in antmaze tasks. The improvements of our DiffCPS compared to these algorithms are shown in the table below.
>
> | D4RL Tasks              | DiffusionQL (ICLR 2023) | SfBC (ICLR 2023)   |IQL (ICLR 2022)|
> |-------------------------|-------------------|-------------|-----------|
> | Locomotion Average       | 5%           | 22%       |20%|
> | AntMaze Average          | 17%            | 10%      |30%|
>
> **Table 2:** The improvements of our DiffCPS compared to recent ICLR offline RL algorithms.
>
> Compared with other offline reinforcement learning algorithms, it's evident that our algorithm exhibits a significant improvement over the baseline.
>
> >   **Q4:** In the toy example, the authors claim that "SfBC incorrectly models the dataset as a circle instead of a noisy circle". I checked the image and did not find such a difference. Can the authors give an explanation?
>
> **A4:** Thanks for pointing this out. We've realized that our statement might lead to misunderstandings. We've revised the statement in our revision. What we intended to convey is that due to the architecture of SfBC introducing unnecessary sampling errors at T=15, it generates points that differ significantly from the dataset.

---

> > ### Author Response · Authors · 2023-11-21
> > **Response Continued**
> >
> > >  **Q5:** The authors deploy diffusion-based policies in place of the traditional Gaussian policies to handle the multi-modal problem. I wonder whether there are choices other than diffusion models and naive Gaussians. For example, flow models can express a rich distribution class and have explicit densities. Is it possible to fit the flow models into the framework of AWR?
> >
> > **A5:** In fact, it is possible to incorporate a flow-based model into AWR, but our theoretical framework does not apply to flow-based models. This is because the joint distribution $\mu(a^{0:T}\vert s)$ defined by Equation 9 is not always present for flow-based models. In contrast, diffusion models generally define the joint distribution $\mu(a^{0:T}\vert s)$. As a result, the policy constraint in Equation 8 cannot be eliminated for flow-based models, rendering our theoretical framework inapplicable to them.
> >
> > [1] RT Rockafellar. Convex analysis. Princeton Math. Series, 28, 1970.
> >
> > [2] Paternain, Santiago, Luiz F. O. Chamon, Miguel Calvo-Fullana, and Alejandro Ribeiro. “Constrained Reinforcement Learning Has Zero Duality Gap.” arXiv, October 29, 2019. https://doi.org/10.48550/arXiv.1910.13393.
> >
> > [3] Haarnoja, Tuomas, Aurick Zhou, Pieter Abbeel, and Sergey Levine. “Soft Actor-Critic: Off-Policy Maximum Entropy Deep Reinforcement Learning with a Stochastic Actor.” arXiv, August 8, 2018. https://doi.org/10.48550/arXiv.1801.01290.
> >
> > [4] Chen, Huayu, Cheng Lu, Chengyang Ying, Hang Su, and Jun Zhu. “Offline Reinforcement Learning via High-Fidelity Generative Behavior Modeling.” arXiv Preprint arXiv:2209.14548, 2022.
> >
> > [5] Kostrikov, Ilya, Ashvin Nair, and Sergey Levine. “Offline Reinforcement Learning with Implicit Q-Learning.” arXiv, October 12, 2021. http://arxiv.org/abs/2110.06169.
> >
> > [6] Wang, Zhendong, Jonathan J. Hunt, and Mingyuan Zhou. “Diffusion Policies as an Expressive Policy Class for Offline Reinforcement Learning.” arXiv Preprint arXiv:2208.06193, 2022.

---

> ### Comment · Reviewer_4x5N · 2023-11-22
>
> Thanks for the authors' response. I find the revised version more readable. The revised proof appears to be sound. Nevertheless, I still have several questions after checking the proof in detail.
>
> 1, **About equation (34).** In equation (34), the authors use the relationship $\rho^{\pi_b}(s)\approx\rho^{\mu_t}(s)\approx\rho^{\mu_1}(s)\approx\rho^{\mu_2}(s)$. What does the $\approx$ mean here? Why does such a relationship hold?
>
> 2, **About the Slater's condition.** When verifying the Slater's condition, the authors only verify that for a specific $\kappa_0$ the Slater's condition holds. My interpretation is that this means the Slater's condition might only be satisfied for certain choices of $\epsilon$. Am I correct? However, it seems the authors conclude that the Slater's condition holds for all $\epsilon$.
>
> If the authors could address my concerns I would consider raising my score.
>
> (*Some random thoughts*: I understand this paper may not be a theory-oriented work. And I agree that intuitively it is totally fine to apply the dual methods even if the strong duality is only "approximately true". But I think if the authors choose to state their intuitions in the form of mathematical theorems, then they should ensure full technical rigor. )

---

> > ### Author Response · Authors · 2023-11-22
> > **Response to 4X5N**
> >
> > We thank reviewer 4X5N for your prompt response. **We have updated the proof in Appendix A.2  to address your concerns (in blue).** Back to your concerns on $\rho^{\pi_b}(s)\approx\rho^{\mu_t}(s)\approx\rho^{\mu_2}(s)$  and Slater's conditions.
> >
> > > Question about $\rho^{\pi_b}(s)\approx\rho^{\mu_t}(s)\approx\rho^{\mu_2}(s)$
> >
> > Thank you for pointing that out. Because all feasible policies satisfy the KL divergence constraint, when the KL divergence constraint is sufficiently strong, the differences between the occupancy measures for different policies are small. Therefore, based on equation 26, we can assume that for all feasible policies $\rho^{\pi_b}(s)\approx\rho^{\mu_1}(s)\approx\rho^{\mu_2}(s)$. We recognize that our previous statement may lead to ambiguity. Therefore, in the revised version, we have placed it before the lemma A.1. First, we explain the simplified constraint form (equation 28) and its rationale (equation 26). Then, in the proof, we establish that the perturbation problem  (equation 30) for the simplified optimization problem is concave. Finally, we prove the strong duality based on the lemma.
> > > About the Slater's condition.
> >
> > Thank you for pointing that out. We have modified the description of the theorem. Following  Santiago [1], we introduced Slater's condition as a prerequisite for Theorem 3.2. Our previous explanation of Slater's condition was meant to emphasize that satisfying Slater's condition is a relatively mild condition for the constraint problems in equations 6-8, since for all  $\kappa_0>-E_{s\sim \rho^{\pi_b}(s),a\sim\pi_b(\cdot\vert s)}[\log\pi_b(\cdot\vert s)]$, we can get $\mu=\pi_b$ is strictly feasible, which meets Slater's conditions.
> >
> >
> > [1] Paternain, Santiago, Luiz F. O. Chamon, Miguel Calvo-Fullana, and Alejandro Ribeiro. “Constrained Reinforcement Learning Has Zero Duality Gap.” arXiv, October 29, 2019. https://doi.org/10.48550/arXiv.1910.13393.

---

### Official Review · Reviewer_hHF8 · 2023-10-30

**Soundness:** 4 excellent
**Presentation:** 4 excellent
**Contribution:** 3 good
**Rating:** 8
**Confidence:** 4

**Summary:**

The authors tackle a KL-constrained offline RL problem, where a RL policy is trained but additional constraints are put on the distribution shift between the original policy and the trained policy. The authors point out the weakness in using a unimodal Gaussian policy and propose to use a diffusion process as a policy parametrization. The authors show some desirable properties of using diffusion as a policy parametrization, and propose a primal-dual iteration algorithm to solve KL-constrained offline RL. The authors compare against baselines in D4RL and present ablation studies of the algorithm.

**Strengths:**

1. Overall, this is a good paper with strong theoretical and empirical results.
2. The visualization for the problem of having unimodal policies is very intuitive, and motivates a richer class of policies that can model more multimodal distributions in offline RL.
3. The ablation studies of the algorithm is extensive and the authors have done good research into best practices in training diffusion models.

**Weaknesses:**

1. The proposed baselines are entirely numerical and quantitative; while convincing, it would have been nice to see some qualitative behavior of DiffCPS compared to other baselines as well in order to strengthen the authors' claims. For instance, trajectory stitching is a popular example in offline RL where unimodal policies might possibly fail if done naively. In simple offline RL tasks such as the X data collection task in Diffuser [1], does DiffCPS succeed?

2. One interpretation of the authors' work is that when we use a generative model of the training data from the behavior policy, this has the effect of automatically constraining the distribution shift between the learned policy and the behavior policy. The authors are missing some relevant work in this direction in the context of model-based offline RL. For instance, [2] motivates a very similar objective with DiffCPS, where the cross-entropy term is penalized rather than constrained (with the difference that the distribution shift is on the state-action occupation measures rather than the action distribution in the model-based setting).

3. On a related note, offline RL also has model-free and model-based approaches, and the author's approach is model-free. The title might give the impression that this method is model-based, though I am aware that the authors' intention was to say it's based on a diffusion generative model. Maybe Diffusion based might be a better title?

[1] Janner et al., "Planning with Diffusion for Flexible Behavior Synthesis", ICML 2022

[2] Suh et al., "Fighting Uncertainty with Gradients: Offline Reinforcement Learning with Diffusion Score Matching", CoRL 2023

**Questions:**

1. Have the authors considered using Augmented Lagrangian?
2. In Theorem 3.1, what is $d_{\pi_b(s)}$? Should this be the occupation measure of the behavior policy, which the authors previously denote using $\rho$? or the initial distribution of the behavior policy?
3. How specific are the authors' claims to diffusion models? For instance, if we had trained a denoising autoencoder model (assuming they can be trained well), would the theorems still hold?

---

> ### Author Response · Authors · 2023-11-21
> **Response to hHF8**
>
> We thank reviewer hHF8 for providing positive feedback and helpful suggestions. Below please find a response.
>
> >  **Q1:** Qualitative behavior of DiffCPS compared to other baselines
>
> **A1:**
> Thanks for suggesting this. We have added an experiment to show the behavior of DiffCPS in Appendix F.1
>
> >  **Q2:** missing some relevant work
>
>   **A2:** Thanks for pointing that out. We have supplemented information about the related work on model-based offline reinforcement learning in Appendix E. And we will emphasize here the distinctions between our approach and [1]. [1] introduces a method that utilizes the score function of the Diffusion Model to approximate the gradient of the proposed uncertainty estimation. Our method, DiffCPS, diverges from this approach in several key aspects:
> (i) DiffCPS tackles the constrained policy search problem in offline RL through Lagrange dual and recursive optimization, while the method in the paper employs first-order optimizers directly.
> (ii) DiffCPS is a model-free method, in contrast to the model-based nature of the method in [1].
> (iii) The objective function in DiffCPS differs as it incorporates the loss from DDPM instead of score matching, and our coefficient is auto-adaptive.
> (iv) Our experimental results in D4RL demonstrate the superior performance of DiffCPS over the method in the paper, attributed to the design of our algorithm.
>
> >   **Q3:** Maybe Diffusion based might be a better title?
>
>   **A3:** Thanks for suggesting this. We notice that "diffusion model-based" could be ambiguous to some extent. Following your advice, we have changed the title and algorithm names to Diffusion-based constrained policy search to eliminate ambiguity.
>
> >   **Q4:** Have the authors considered using the Augmented Lagrangian Method (ALM)?
>
>   **A4:** Initially, for simplicity, we directly applied the alternating optimization (Dual Ascent) of SAC [2] to solve the dual problem. Theoretically, using ALM could offer better convergence performance. Our experiments on HalfCheetah demonstrate that using ALM  increases training stability. However, the final score is slightly lower than the Dual Ascent method. This could be attributed to (i) ALM introducing penalty terms, making the optimal target KL divergence for Dual Ascent not necessarily optimal for ALM, and (ii) the penalty function making the policy too conservative, thereby preventing the adoption of high-reward actions. We plan to further explore the use of ALM to optimize our problem in future research. You can find our discussion on the application of ALM in our algorithm in Appendix G.
>
> >  **Q5:**  question about $d_{\pi_b(s)}$
>
>   **A5:** Thanks for pointing this out. $d_{\pi_b(s)}$ denotes the unnormalized discounted state visitation frequencies of the behavior policy. We acknowledge the issues with the representation of some notations in the paper. In the revised version, we use $\rho^{\pi_b}(s)$ to represent the unnormalized discounted state visitation frequencies of the behavior policy and provide a notation table in Appendix B.
>
> >   **Q6:** How specific are the authors' claims to diffusion models? For instance, if we had trained a denoising autoencoder model (assuming they can be trained well), would the theorems still hold?
>
>   **A6:** In fact, when using a denoising autoencoder instead of the diffusion model, the theoretical properties of DiffCPS are no longer guaranteed. This is because (i) Constraint $\int_a \mu(a\vert s)da =1$  is challenging to establish for other models; it holds for the diffusion model because $\mu(a\vert s) = \int \mu(a^{0:T}\vert s)da^{1:T}$ holds. (ii) Denoising autoencoders struggle to accurately approximate the constraint on KL divergence.
>
>
> [1]  Suh et al., "Fighting Uncertainty with Gradients: Offline Reinforcement Learning with Diffusion Score Matching", CoRL 2023
>
> [2] Haarnoja, Tuomas, Aurick Zhou, Pieter Abbeel, and Sergey Levine. “Soft Actor-Critic: Off-Policy Maximum Entropy Deep Reinforcement Learning with a Stochastic Actor.” arXiv, August 8, 2018. https://doi.org/10.48550/arXiv.1801.01290.

---

> ### Comment · Reviewer_hHF8 · 2023-11-23
> **comment**
>
> I would like to thank the authors for the detailed response, I mostly agree with the points and will keep the score as is.

---

### Official Review · Reviewer_UbgT · 2023-11-03

**Soundness:** 2 fair
**Presentation:** 3 good
**Contribution:** 2 fair
**Rating:** 5
**Confidence:** 4

**Summary:**

The paper studies a constrained policy search in offline reinforcement learning. To increase the expressivity of Gaussian-based policies, the authors propose to use diffusion model to represent policy. The authors formulate a diffusion model based constrained policy optimization problem, and propose a constrained policy search algorithm. The authors also provide experiments to show the performance of this method.

**Strengths:**

- The paper is well organized, and the key idea is delivered.

- The authors provide an example to show the expressivity limitation in standard advantage regression methods, which justifies the necessity of introducing diffusion models.

- The authors use a popular diffusion model: DDPM to represent policy, and present a new constrained policy search method, which is intuitively simple and easy to implement.

- Experimental results demonstrate comprable performance compared with state-of-the-art methods.

**Weaknesses:**

- The importance or motivation of propositions, theorems, and corollaries is not well explained. Most results can be more directly obtained using simple calculations that are known in diffusion model.

- The use of diffusion model as policy in constrained policy search has been studied in offline RL. Due to the similarity, it is important to distinguish them in an explicit way.

- The authors claim strong duality for Equation (12) according to the duality in the convex optimization. However, the constrained policy search is a non-convex problem. It is not justified if the strong duality still holds.

- The main result is empirical. It is useful if the authors could provide performance analyses, which can strengthen the method with solid theoretical guarantees.

**Questions:**

- A large paragraph of this paper introduces known results. Can the authors highlight more new developments compared with existing methods?

- It is not clear to me the strong duality of Equation (12). Can the authors justify it?

- How does the problem (18)-(19) can solve the original problem (12)?

- Are there other diffusion models useful for constrained policy search? It is useful if the authors could discuss the generalization of this approach.

- Training diffusion model can be inefficient. What are computational times for the methods in experiments?

- The examples in experiments are created in simulated environments. Any realistic offline dataset you can use to show performance of your algorithm?

---

> ### Author Response · Authors · 2023-11-21
> **Response to UbgT**
>
> We thank reviewer UbgT for the comments and suggestions. Below, we address the concerns raised in your review point by point. Please let us know if you have any further concerns or whether this adequately addresses all the issues that you raised with the paper.
> >   **Q1:** The importance or motivation of propositions, theorems, and corollaries is not well explained. Most results can be more directly obtained using simple calculations that are known in the diffusion model.
>
>   **A1:** In fact, the logic of our theorems, proposition, and corollary is as follows: Theorem 3.1 and Corollary 3.1.1 establish conditions for Theorem 3.2 to hold. Theorem 3.2 proves the strong duality for the constrained policy search problem with policies based on the diffusion model. This allows us to use the dual problem to find the optimal solution (optimal policy) for the original problem (Theorem 3.2 in the revised paper). Proposition 3.1 is introduced for solving the dual problem, stating that the loss of DDPM can serve as an approximation to $H(\pi_b,\mu)$. In the revised version, we have reorganized the theorems, proposition, and collary for a clearer presentation.
>
> >  **Q2:** A large paragraph of this paper introduces known results. Can the authors highlight more new developments compared with existing methods?
>
>   **A2:** Compared to other methods using the diffusion model, we directly employ diffusion-based policy to solve constrained policy search. We demonstrate the previously unexplored strong duality for the constrained policy search problem with policies based on the diffusion model. Our research indicates that the favorable properties of the diffusion model can be leveraged for constrained policy search in offline RL, addressing the limited expressivity of Gaussian-based models.
>
> >   **Q3:** It is not clear to me the strong duality of Equation (12). Can the authors justify it?
>
>   **A3:** We have rigorously re-proven the strong duality of Equation 12 in Appendix A.2. New proof relies on a well-known result from perturbation theory connecting strong duality to the convexity of the perturbation function. Here, we formally state the lemma used in the proof, which is referred to [1] Cor. 30.2.2 and [2] Ch 3.
>
> >If (i) $r$ is bounded; (ii) Slater's condition holds for the diffusion-based constrained policy search and (iii) its perturbation function~$P(\xi)$ is concave, then strong duality holds for equation 12.
>
> The proof consists of two main steps (See details in Appendix A.2. in our revision): (i) demonstrating that the perturbation problem of the simplified diffusion-based constrained search problem, as per Theorem 3.1, is a concave function, and (ii) proving the Slater's condition holds for the diffusion-based constrained search problem. And we have also rephrased the main theorems of the paper to ensure a more coherent logical flow.
>
> >   **Q4:** How does the problem (18)-(19) can solve the original problem (12)?
>
>   **A4:** The dual function for Problem 12 is given by Equation 14. Following SAC[3], we can use alternating optimization to solve the dual problem. Theorem 3.2 also proves that the policy obtained through the dual problem is optimal for the original problem. In fact, Equation 18 and Equation 19 represent the objective functions for the policy and Lagrange multipliers. By applying stochastic gradient descent to Equation 18 and Equation 19, we can approximately solve the dual problem Equation 14, and consequently address Problem 12. The updated paper includes a clearer pseudocode (Ch3) for DiffCPS, highlighting this process.
>
> >   **Q5:** Are there other diffusion models useful for constrained policy search? It is useful if the authors could discuss the generalization of this approach.
>
>   **A5:** Other methods based on the diffusion model can also leverage our framework. This is because other diffusion model-based methods, such as Score-Based Generative Models (SGMs) and Stochastic Differential Equations (Score SDEs), define a common distribution for all actions. Thus, $\mu(a\vert s) = \int \mu(a^{0:T}\vert s)da^{1:T}$ holds for other diffusion model-based methods, allowing the elimination of constraints $\int_a \mu(a\vert s)da =1$  and the application of our theoretical framework.

---

> > ### Author Response · Authors · 2023-11-21
> > **Response Continued**
> >
> > >  **Q6:**  Training diffusion model can be inefficient. What are the computational times for the methods in experiments?
> >
> >   **A6:** Thanks for suggesting this. We have included the algorithm's runtime in Appendix C. The table below displays the training time for DiffCPS. The runtime of DiffCPS is comparable to other diffusion model-based methods, and further incorporating acceleration techniques from other diffusion models can enhance execution speed.
> > | D4RL Tasks              | DiffCPS (T=5) | DiffusionQL (T=5) | SfBC (T=5)  |
> > |-------------------------|---------------|-------------------|-------------|
> > | Locomotion Runtime (1 epoch)      | 7.68s         | 5.1s              | 8.42s       |
> > | AntMaze Runtime  (1 epoch)       | 9.96s         | 10.5s             | 10.53s      |
> >
> > **Table 1:** Runtime of different diffusion-based offline RL methods.
> >
> >
> >
> > >   **Q7:** The examples in experiments are created in simulated environments. Any realistic offline dataset you can use to show performance of your algorithm?
> >
> >   **A7:** We conducted preliminary experiments on the TriFinger dataset [4], when trained with 2e6 training steps (gradient steps) on the sim-expert dataset, DiffCPS (T=45) achieves a success rate of $0.906\pm0.001$ in the push task based on 100 evaluation episodes and 5 random seeds, as detailed in table 2.
> >   | TriFinger-Push          | BC            | CRR               | IQL             | DiffCPS (Ours)            |
> > |-------------------------|---------------|-------------------|-----------------|---------------------------|
> > | Sim-Expert              | $0.83\pm0.02$ | $\bf{0.94\pm0.04}$| $0.88\pm0.04$   | $\bf{0.906\pm0.001}$      |
> >
> > **Table 2:** The success rate of DiffCPS and other baselines on the TriFinger Push task.
> >
> >   These results indicate that  DiffCPS's performance matches the baseline algorithms. It's worth noting that our DiffCPS still utilizes a simple MLP to represent the policy, suggesting potential benefits from more powerful network architectures. However, training on real datasets is time-consuming, even for carefully accelerated diffusion model-based offline reinforcement learning methods. Therefore, we focused on partial evaluations for specific tasks rather than completing the entire assessment on the TriFinger dataset.
> >
> >
> >
> > [1] RT Rockafellar. Convex analysis. Princeton Math. Series, 28, 1970.
> >
> > [2] Paternain, Santiago, Luiz F. O. Chamon, Miguel Calvo-Fullana, and Alejandro Ribeiro. “Constrained Reinforcement Learning Has Zero Duality Gap.” arXiv, October 29, 2019. https://doi.org/10.48550/arXiv.1910.13393.
> >
> > [3] Haarnoja, Tuomas, Aurick Zhou, Pieter Abbeel, and Sergey Levine. “Soft Actor-Critic: Off-Policy Maximum Entropy Deep Reinforcement Learning with a Stochastic Actor.” arXiv, August 8, 2018. https://doi.org/10.48550/arXiv.1801.01290.
> >
> > [4] Gürtler, Nico, Sebastian Blaes, Pavel Kolev, Felix Widmaier, Manuel Wüthrich, Stefan Bauer, Bernhard Schölkopf, and Georg Martius. “BENCHMARKING OFFLINE REINFORCEMENT LEARNING ON REAL-ROBOT HARDWARE,” 2023.

---

> > ### Comment · Reviewer_UbgT · 2023-11-22
> >
> > Thank you for the response. I have a few further questions.
> >
> > For Q3, does there exist a weaker condition for strong duality than Slater condition? It looks that the constraint function in Problem 12 is strictly convex. Does this help getting tighter duality?
> >
> > For Q4, it is not very clear to me how the stochastic gradient updates for Equation 18 and Equation 19 can solve Problem 12. It is known that it is not true even for a linear program. Aslo, Algorithm 1 has missed the constraint sets.

---

> > > ### Author Response · Authors · 2023-11-23
> > > **Response to UbgT**
> > >
> > > We thank reviewer UbgT for your response.  Back to your concerns on Q3 and Q4.
> > >
> > > > For Q3, does there exist a weaker condition for strong duality than the Slater condition? It looks like the constraint function in Problem 12 is strictly convex. Does this help get tighter duality?
> > >
> > > Our updated proof is based on the policy's occupancy measure, requiring the use of Slater's condition to establish the validity of Lemma A.1. However, Slater's condition is easily satisfied in the context of the constraint problem in Equation 12. For all $\kappa_0 > -E_{s\sim \rho^{\pi_b}(s),a\sim\pi_b(\cdot\vert s)}[\log\pi_b(\cdot\vert s)]$, we can obtain $\mu=\pi_b$ is strictly feasible (since KL divergence is non-negative, $\kappa_0 > -E_{s\sim \rho^{\pi_b}(s),a\sim\pi_b(\cdot\vert s)}[\log\pi_b(\cdot\vert s)]$ implies $\epsilon> 0$ in the constraint), which satisfies Slater's conditions.
> > >
> > > > For Q4, it is not very clear to me how the stochastic gradient updates for Equation 18 and Equation 19 can solve Problem 12. It is known that it is not true even for a linear program. Also, Algorithm 1 has missed the constraint sets.
> > >
> > > We follow SAC's [1] approach of alternating optimization to **approximately solve** the dual problem in Equation 14. Specifically, we optimize the policy network based on Equation 20, then fix the policy and optimize the Lagrange multiplier using Equation 19. We repeat this process to **approximately solve** the dual problem 14. Before Equation 20, we also mentioned our method of constraining lambda to be non-negative. We  cliped the $\lambda$ to keep the constraint $\lambda \geq 0$ holding by $\lambda_\text{clip}=\max(c,\lambda)$, $c\geq0$. However, in Algorithm 1, we forgot to explicitly state this, and we appreciate your pointing out the error in Algorithm 1. We have also updated Algorithm 1 to correct this error.
> > >
> > > [1] Haarnoja, Tuomas, Aurick Zhou, Pieter Abbeel, and Sergey Levine. “Soft Actor-Critic: Off-Policy Maximum Entropy Deep Reinforcement Learning with a Stochastic Actor.” arXiv, August 8, 2018. https://doi.org/10.48550/arXiv.1801.01290.

---

### Author Response · Authors · 2023-11-21
**Response to All**

We thank all the reviewers for the time and expertise they have invested in these reviews and for their positive and constructive feedback. Your comments and suggestions have helped us to improve the paper. We provide a response and clarifications below for each reviewer respectively and hope they can address your concerns.

We’ve also updated the paper with a few modifications to address reviewer suggestions and concerns (in blue). Summary of updates below:

- We have corrected some errors in notations and expressions in the paper.
- We have reorganized the descriptions of the theorems and algorithms to enhance the clarity of the paper.
- We have rigorously re-proven the strong duality of Equation 12 in Appendix A.2.
- We added a Notation table in Appendix B.
- We added the runtime of DiffCPS in Appendix C.
- We added the extra related work about model-based offline RL in Appendix E.
- We add an experiment to show the behavior of DiffCPS in Appendix F.1.
- We added a  realistic offline dataset experiment in Appendix F.2.
- We added a discussion of the Augmented Lagrangian Method in Appendix G.

---

### Meta-Review · Area_Chair_LxuY · 2023-12-10

**Metareview:**

The paper tackles the challenge of constrained policy search in offline reinforcement learning. It highlights the limitations of using unimodal Gaussian policies and suggests employing a diffusion process for increased expressivity. The proposed algorithm, DiffCPS, is presented as a solution to the KL-constrained offline RL problem, avoiding the density calculation challenges of the traditional AWR framework. The paper includes theoretical justification for the diffusion-based policy and conducts an empirical study, comparing DiffCPS against baselines using the D4RL benchmark. The results indicate the efficacy of DiffCPS, providing a promising alternative in addressing offline RL problems.

The paper provides both theory and empirical results, offering clear and well-organized insights. It effectively illustrates the limitations of standard methods, advocating for the use of diffusion models for richer expressivity. The proposed algorithm is intuitively simple, leveraging a popular diffusion model (DDPM) and demonstrating comparable performance with state-of-the-art methods. The visualization for policy limitations is intuitive, motivating the exploration of more diverse policy classes. Thorough ablation studies and research into diffusion model training best practices enhance the paper's overall quality. The inclusion of a well-designed toy example and numerical experiments further strengthens its impact.

On the other hand, the paper faces significant theoretical flaws in key propositions, compromising the validity of duality results. Clarity is lacking in notations and descriptions, potentially confusing readers. Empirical improvements of the proposed algorithm are marginal, and the absence of a qualitative comparison weakens the claims. Relevant work in model-based offline RL is overlooked, and there's an inconsistency in claiming strong duality for a non-convex problem.

The authors' rebuttals have solved some of the concerns raised by the reviewers, particularly those related to the clarity of the presentations. However, the issues related to the soundness of the theoretical part make the paper not ready for publication.
We encourage the authors to consider the reviewers' suggestions while preparing a new version of their paper.

**Justification For Why Not Higher Score:**

The paper presents critical theoretical flaws, unclear notations, marginal empirical improvements, and oversight of relevant work.

**Justification For Why Not Lower Score:**

N/A

---

### Decision · Program_Chairs · 2024-01-16

Reject